# The Na$^+$/K$^+$ pump dominates control of glycolysis in hippocampal dentate granule cells

Dylan J Meyer, Carlos Manlio Díaz-García[†], Nidhi Nathwani, Mahia Rahman, Gary Yellen*

Department of Neurobiology, Harvard Medical School, Boston, United States

**Abstract** Cellular ATP that is consumed to perform energetically expensive tasks must be replenished by new ATP through the activation of metabolism. Neuronal stimulation, an energetically demanding process, transiently activates aerobic glycolysis, but the precise mechanism underlying this glycolysis activation has not been determined. We previously showed that neuronal glycolysis is correlated with Ca$^{2+}$ influx, but is not activated by feedforward Ca$^{2+}$ signaling (Díaz-García et al., 2021a). Since ATP-powered Na$^+$ and Ca$^{2+}$ pumping activities are increased following stimulation to restore ion gradients and are estimated to consume most neuronal ATP, we aimed to determine if they are coupled to neuronal glycolysis activation. By using two-photon imaging of fluorescent biosensors and dyes in dentate granule cell somas of acute mouse hippocampal slices, we observed that production of cytoplasmic NADH, a byproduct of glycolysis, is strongly coupled to changes in intracellular Na$^+$, while intracellular Ca$^{2+}$ could only increase NADH production if both forward Na$^+$/Ca$^{2+}$ exchange and Na$^+$/K$^+$ pump activity were intact. Additionally, antidromic stimulation-induced intracellular [Na$^+$] increases were reduced >50% by blocking Ca$^{2+}$ entry. These results indicate that neuronal glycolysis activation is predominantly a response to an increase in activity of the Na$^+$/K$^+$ pump, which is strongly potentiated by Na$^+$ influx through the Na$^+$/Ca$^{2+}$ exchanger during extrusion of Ca$^{2+}$ following stimulation.

*For correspondence: gary_yellen@hms.harvard.edu

Present address: [†]Department of Biochemistry and Molecular Biology, University of Oklahoma Health Sciences Center, Oklahoma City, United States

## Editor's evaluation

The authors investigate mechanisms that regulate glycolytic ATP production in neurons. They conclude that the cytosolic Na$^+$, not Ca$^{2+}$, and the activity of the Na$^+$/K$^+$ pump drive glycolysis. The study is conceptually significant as it seeks to determine how neuronal glycolysis is coupled to electrical activity. The study is thoughtful, uses sophisticated fluorescence lifetime imaging technology and clever experimental designs, and as such provides new insights into how electrical activity regulates glycolysis.

## Introduction

Cellular energy consumption must be balanced by new energy production through the activation of metabolism. In neurons, energy consumption is strongest following excitation (**Attwell and Iadecola, 2002**; **Attwell and Laughlin, 2001**; **Engl and Attwell, 2015**; **Howarth et al., 2012**; **Lennie, 2003**; **Yu et al., 2018**). Accordingly, neuronal excitation triggers a transient activation of glycolysis, which is measured as an increase to neuronal cytoplasmic NADH:NAD$^+$ (NADH$_{CYT}$) that can last for several minutes after the excitatory event (**Díaz-García et al., 2017**; **Díaz-García et al., 2021a**). The specific mechanism that drives neuronal glycolysis activation has not been identified.

It seems highly plausible that glycolysis activation in neurons could be driven primarily by an increase in the activity of ion-ATPases, the ATP-powered ion transporters that move ions against their electrochemical gradients by (through a series of reaction steps; *Post et al., 1965*; *Sen and Post, 1964*) converting ATP into ADP and $P_i$ (*Biondo et al., 2021*; *Calì et al., 2018*; *Chemaly et al., 2018*; *Kaplan, 2002*). Ion-ATPases are estimated to be responsible for most of the energy consumption associated with neuronal signaling and are closely coupled to glycolytic enzymes across many cell types (*Ames, 2000*; *Dhar-Chowdhury et al., 2007*; *Dzeja and Terzic, 2003*; *Xu et al., 1995*). Moreover, activation of glycolysis by increased ion-ATPase activity could be multimodal: a decrease to cellular ATP would stimulate glycolysis by mass action at the ATP-generating steps, while an increase to cellular ADP and $P_i$ would stimulate glycolysis by allosterically activating phosphofructokinase and by mass action at both the glyceraldehyde-3-phosphate dehydrogenase/phosphoglycerate kinase enzyme complex and pyruvate kinase steps (*Fothergill-Gilmore and Michels, 1993*; *Kemp and Foe, 1983*; *Nelson et al., 2008*; *Schöneberg et al., 2013*; *Tomokuni et al., 2010*).

There are several ion-ATPases (also called ion pumps) that regulate neuronal $Na^+$ and $Ca^{2+}$ changes (*Figure 1A*). $Na^+$ is handled exclusively by the $Na^+/K^+$-ATPase (or $Na^+/K^+$ pump), which uses 1 ATP molecule to export 3 $Na^+$ and import 2 $K^+$ across the plasma membrane. $Ca^{2+}$ is handled by a diverse set of active transporters, including the plasma membrane $Ca^{2+}$-ATPase (PMCA), which exports 1 $Ca^{2+}$ across the plasma membrane per ATP (*Niggli et al., 1982*; *Thomas, 2009*), and the sarco-/endoplasmic reticulum $Ca^{2+}$-ATPase (SERCA), which pumps 2 $Ca^{2+}$ into the endoplasmic reticulum lumen per ATP (*Tran et al., 2009*). Aside from ATPases, $Ca^{2+}$ is also regulated by the $Na^+/Ca^{2+}$-exchanger (NCX) (*Lee et al., 2009*), a secondary active transporter that uses the electrochemical energy stored in the plasma membrane $Na^+$ gradient (built by the $Na^+/K^+$ pump) to actively export 1 $Ca^{2+}$ by importing 3 $Na^+$. Any increase to neuronal $Na^+$ or $Ca^{2+}$ will increase the activity of their respective ion transporters.

Transient increases to neuronal cytoplasmic $Ca^{2+}$ ($Ca^{2+}_{CYT}$) induced by excitation are positively correlated with transient increases in subsequent $NADH_{CYT}$ production (*Díaz-García et al., 2017*; *Díaz-García et al., 2021a*); in other words, stronger stimulations evoke larger increases to both $Ca^{2+}_{CYT}$ and $NADH_{CYT}$. At first glance, this positive correlation could be (mis)interpreted as glycolysis being driven primarily by an increase in the activities of $Ca^{2+}$ pumps (PMCA and SERCA) as $Ca^{2+}$ is pumped from the cytoplasm. However, this readout of neuronal $NADH_{CYT}$ and $Ca^{2+}_{CYT}$ lacks information about stimulation-induced increases to intracellular $Na^+$, which also likely covary with stimulation strength, so a contribution of increased $Na^+/K^+$ pump activity to glycolysis activation cannot be excluded.

The $Na^+/K^+$ pump is primed to be the predominant driving force underlying neuronal glycolysis activation. Many reports estimate that the $Na^+/K^+$ pump consumes ~50% of total brain energy (*Ames, 2000*; *Astrup et al., 1981*; *Engl and Attwell, 2015*; *Milligan and McBride, 1985*; *Whittam, 1962*) and uses substantially more ATP than $Ca^{2+}$ pumps both during neurotransmission and at rest (*Attwell and Laughlin, 2001*; *Harris et al., 2012*; *Rolfe and Brown, 1997*). Furthermore, since $Ca^{2+}_{CYT}$ regulation in some neurons depends strongly on the NCX (*Lee et al., 2009*), a transient $Ca^{2+}_{CYT}$ increase could indirectly elevate intracellular $[Na^+]$, which would potentiate any increase in $Na^+/K^+$ pump activity due to channel-mediated $Na^+$ entry.

To shed light on the mechanism of neuronal glycolysis activation, we investigated the coupling of $Na^+/K^+$ pump and $Ca^{2+}$ pump activities to $NADH_{CYT}$ production in the somas of hippocampal dentate granule cells (DGCs) within acute brain slices by using two-photon fluorescence imaging of genetically encoded fluorescent biosensors and fluorescent dyes. The data provide compelling evidence indicating that increased activity of the $Na^+/K^+$ pump is the predominant driver of neuronal glycolysis activation, whereas increased activities of $Ca^{2+}$ pumps appear to be negligibly coupled to glycolysis.

## Results

### Cytoplasmic NADH production is strongly influenced by Na⁺, but not Ca²⁺

We simultaneously monitored changes in neuronal metabolism and ion fluxes in DGC somas by expressing Peredox and RCaMP, two genetically encoded fluorescent biosensors that report $NADH:NAD^+$ ($NADH_{CYT}$) and free $[Ca^{2+}]$, respectively (*Akerboom et al., 2013*; *Hung et al., 2011*; *Mongeon et al., 2016*). Analyte binding to Peredox (NADH in competition with $NAD^+$) or RCaMP ($Ca^{2+}$) alters each sensor's fluorescence lifetime (LT), the average time that the fluorophore spends

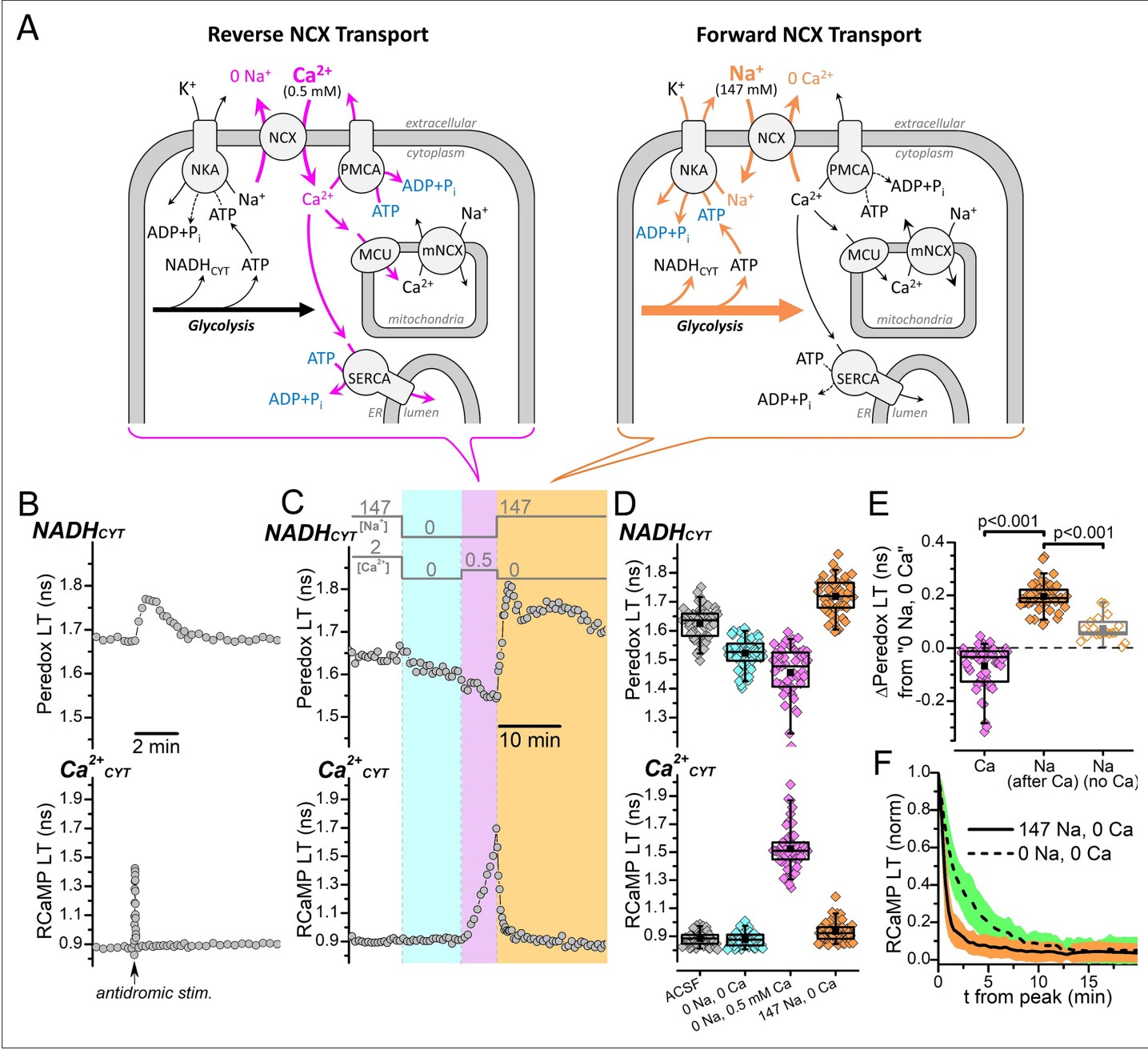

**Figure 1.** NADH$_{CYT}$ production is strongly influenced by Na$^+$, but not by Ca$^{2+}$$_{CYT}$. (**A**) Cartoon showing the NCX transport modes activated by different external [Na$^+$] and [Ca$^{2+}$] conditions and their expected effects on the activities of ion pumps and the production of NADH$_{CYT}$ from glycolysis activation. Reverse NCX transport (*left schematic*) increases intracellular [Ca$^{2+}$], which increases the activities Ca$^{2+}$ pumps and Ca$^{2+}$ transport into mitochondria (*magenta arrows*). Forward NCX transport (*right schematic*) increases intracellular [Na$^+$], which increases the activity of the Na$^+$/K$^+$ pump (*orange arrows*). The bracket below each schematic indicates the NCX transport mode activated by the external solution changes in (**C**). Transport stoichiometries are not indicated. Abbreviations: Na$^+$/Ca$^{2+}$-exchanger (NCX), Na$^+$/K$^+$-ATPase (NKA), plasma membrane Ca$^{2+}$-ATPase (PMCA), sarco-/endo-plasmic reticulum Ca$^{2+}$-ATPase (SERCA), mitochondrial Ca$^{2+}$ uniporter (MCU), mitochondrial Na$^+$/Ca$^{2+}$-exchanger (mNCX), endoplasmic reticulum (ER). (**B**) Representative fluorescence lifetime (LT) traces of Peredox (*top trace*) and RCaMP (*bottom trace*) from a DGC bathed in ACSF. Antidromic stimulation was delivered at the time point indicated by the arrow along the RCaMP trace, which transiently increases both NADH$_{CYT}$ and Ca$^{2+}$$_{CYT}$. (**C**) Fluorescence LT traces of Peredox (*top*) and RCaMP (*bottom*) from a DGC showing how external Na$^+$ and Ca$^{2+}$ changes affect NADH$_{CYT}$ and Ca$^{2+}$$_{CYT}$. The bars above the Peredox trace indicate the external [Na$^+$] and [Ca$^{2+}$]. NADH$_{CYT}$ was decreased by switching the bath solution from ACSF (147 mM Na$^+$ and 2 mM Ca$^{2+}$) to a solution with nominally 0 Na$^+$ and 0 Ca$^{2+}$ (*cyan shading*). Ca$^{2+}$$_{CYT}$ was elevated by applying 0.5 mM Ca$^{2+}$ with 0 Na$^+$ to activate reverse NCX transport (*magenta shading*), and NADH$_{CYT}$ decreased further. NADH$_{CYT}$ was strongly increased after activating forward NCX transport by the subsequent removal of external Ca$^{2+}$ and application of 147 mM Na$^+$ (*orange shading*). (**D**) Box plots of the fluorescence LTs of Peredox (*top*) and RCaMP (*bottom*) showing

*Figure 1 continued on next page*

**eLife** Research advance

Cell Biology | Neuroscience

*Figure 1 continued*

the effects of the external $Na^+$ and $Ca^{2+}$ changes performed in panel C across many DGCs (n=53). The external bath conditions for each box plot are listed at the bottom of the RCaMP plot in chronological order from left to right. The colors of each box plot correspond to the colors indicated in (**C**). The mean Peredox LT values in each condition were: 1.63±0.06 ns in ACSF, 1.52±0.05 ns in 0 $Na^+$ and 0 $Ca^{2+}$, 1.46±0.09 ns in 0 Na and 0.5 mM $Ca^{2+}$, and 1.72±0.06 ns in 147 mM $Na^+$ and 0 $Ca^{2+}$. The mean RCaMP LT values in each condition were: 0.88±0.05 ns in ACSF, 0.88±0.05 ns in 0 $Na^+$ and 0 $Ca^{2+}$, 1.52±0.15 ns in 0 $Na^+$ and 0.5 mM $Ca^{2+}$, and 0.93±0.07 ns in 147 mM $Na^+$ and 0 $Ca^{2+}$. (**E**) Changes to the Peredox LT relative to the 0 $Na^+$ and 0 $Ca^{2+}$ condition, after either a $Ca^{2+}_{CYT}$ elevation from reverse NCX transport (Ca, black box, magenta filled diamonds), an influx of $Na^+$ due to forward NCX transport (Na after Ca, black box, orange filled diamonds), or application of $Na^+$ without forward NCX (Na no Ca, gray box, orange open diamonds). The mean Peredox LT changes were: –0.07±0.08 ns (n=53) for $Ca^{2+}_{CYT}$ elevation, 0.20±0.05 ns (n=53) for $Na^+$ influx via forward NCX, and 0.07364±0.04565 ns (n=19) for $Na^+$ application without forward NCX. Statistical significance between 'Ca' and 'Na after Ca' is indicated by a paired Wilcoxon test and between 'Na after Ca' and 'Na no Ca' by a Mann-Whitney test. (**F**) Effect of external $Na^+$ on the return of $Ca^{2+}_{CYT}$ to baseline following a reverse NCX transport-mediated $Ca^{2+}$ influx. The mean decay of the RCaMP LT following the $Ca^{2+}_{CYT}$ increase (normalized to the peak RCaMP LT value) is shown when the external solution contained either 147 mM $Na^+$ (*solid line, orange SD shading*, n=53) or 0 $Na^+$ (*dashed line, green SD shading*, n=49). Decay data in 147 mM $Na^+$ were from the same DGCs as in (**D**) and (**E**), while data in 0 $Na^+$ were from the same DGCs as *Figure 1—figure supplement 1B*; a representative trace of this experiment is shown in *Figure 1—figure supplement 1A*.

The online version of this article includes the following source data and figure supplement(s) for figure 1:

**Figure supplement 1.** $NADH_{CYT}$ production does not increase following external $Ca^{2+}$ removal if external $Na^+$ is absent.

**Figure supplement 1—source data 1.** The mean Peredox and RCaMP lifetimes in each external condition for *Figure 1—figure supplement 1B*.

**Figure supplement 2.** Effect of thapsigargin on antidromic stimulation-induced $NADH_{CYT}$ and $Ca^{2+}_{CYT}$ transients.

**Figure supplement 3.** $Ca^{2+}$ entry from reverse NCX transport increases $Ca^{2+}_{MITO}$, which depends on $Na^+$ for removal.

**Figure supplement 3—source data 1.** The mean Peredox and RCaMP lifetimes in each external condition for *Figure 1—figure supplement 3B*.

in the excited state prior to its return to the ground state (*Lakowicz, 2006*). The fluorescence LTs of Peredox and RCaMP are increased (i.e., extended in average duration) by $NADH_{CYT}$ or $Ca^{2+}_{CYT}$ increases, respectively.

Our previous observations demonstrated that $NADH_{CYT}$ increases are strongly tied to glycolysis (*Díaz-García et al., 2017*; *Díaz-García et al., 2021a*) and occur regardless of changes in mitochondrial metabolism: electrical-stimulation-induced increases to $NADH_{CYT}$ were substantially diminished by iodoacetic acid (an inhibitor of GAPDH) but not by AOA (an inhibitor of MAS) and were not affected by impaired NADH production in mitochondria (i.e., in the presence of inhibitors of both the mitochondrial pyruvate carrier and lactate dehydrogenase; Figure 2D in *Díaz-García et al., 2021a*). Thus, acute $NADH_{CYT}$ changes arise directly from changes in glycolytic flux and are not strongly influenced by changes in NADH consumption by the mitochondria.

We began our investigations on the coupling of the $Na^+/K^+$ pump or $Ca^{2+}$ pumps to glycolysis activation by recapitulating our previous findings: that electrical stimulation of DGCs transiently increases their $NADH_{CYT}$ production (*Díaz-García et al., 2017*; *Díaz-García et al., 2021a*). Representative LT traces of Peredox and RCaMP, when both sensors were expressed in the cytoplasm of a DGC within a hippocampal slice that was bathed in ACSF (*Figure 1B*), show that an antidromic stimulation event delivered from an electrode placed in the hippocampal hilus causes both a fast transient increase of the RCaMP LT, reflecting a transient increase to $Ca^{2+}_{CYT}$, and a slower, longer-lasting transient increase of the Peredox LT, reflecting transient overproduction of $NADH_{CYT}$ by glycolysis. But since neuronal stimulation triggers influxes of both $Na^+$ and $Ca^{2+}$, the glycolysis activation could result from increases to both $Na^+/K^+$ pump and $Ca^{2+}$ pump activities.

To differentiate how changes in the activities of the $Na^+/K^+$ pump or $Ca^{2+}$ pumps affect glycolysis, we measured how $NADH_{CYT}$ production is affected by separate elevations of either intracellular $Na^+$ or $Ca^{2+}_{CYT}$ (*Figure 1C*). If glycolysis is preferentially coupled to the activity of either the $Na^+/K^+$ pump or $Ca^{2+}$ pumps, then $NADH_{CYT}$ production should be coupled to changes in the levels of the transported ion ($Na^+$ or $Ca^{2+}$). Simultaneous removal of both $Na^+$ and $Ca^{2+}$ from the bath solution (by external ion substitution) decreased the Peredox LT from its baseline in ACSF (*Figure 1C*), which indicates a decrease to $NADH_{CYT}$; we can attribute this $NADH_{CYT}$ decrease entirely to the removal of $Na^+$, since removing only $Ca^{2+}$ slightly increases $NADH_{CYT}$ (cf. Figure 6—figure supplement 1b in *Díaz-García et al., 2021a*). This $NADH_{CYT}$ decrease suggests that the rate of glycolysis can be slowed by depleting the intracellular $[Na^+]$, which would decrease the activity of the $Na^+/K^+$ pump. The RCaMP LT was not affected by the removal of $Na^+$ and $Ca^{2+}$ (*Figure 1C*), indicating that $Ca^{2+}_{CYT}$ was maintained at or below resting levels, near the floor of RCaMP's dynamic range (*Akerboom et al., 2013*).

Following the removal of external $Na^+$ and $Ca^{2+}$, we added each ion back one at a time to separately increase either intracellular $Na^+$ or $Ca^{2+}$. We increased $Ca^{2+}$ by applying 0.5 mM external $Ca^{2+}$ in the nominal absence of external $Na^+$ to facilitate $Ca^{2+}$ entry via reverse NCX transport, and the expected strong increase in $Ca^{2+}_{CYT}$ was confirmed by an increase in the RCaMP LT (*Figure 1C*, during the magenta shaded interval). However, this increase to $Ca^{2+}_{CYT}$ was not associated with an increase to $NADH_{CYT}$. In fact, the opposite occurred: $Ca^{2+}_{CYT}$ elevation decreased the Peredox LT even further from the 0 $Na^+$ 0 $Ca^{2+}$ bath condition (possibly due to the extrusion of $Na^+$ during reverse NCX; see Discussion). At this point, with $Ca^{2+}_{CYT}$ elevated, we removed 0.5 mM external $Ca^{2+}$ and re-applied external $Na^+$ to facilitate strong $Na^+$ influx via forward NCX transport (in exchange for $Ca^{2+}_{CYT}$), which promptly increased $NADH_{CYT}$. Following the external $Na^+$ application, $Ca^{2+}_{CYT}$ returned to baseline levels (*Figure 1C*, orange shaded interval).

The decrease to Peredox LT by removing external $Na^+$ and the increase to Peredox LT from re-applying external $Na^+$, across many DGCs (*Figure 1D*), show that $NADH_{CYT}$ is strongly influenced by $Na^+$. Relative to the Peredox LT in the 0 $Na^+$ 0 $Ca^{2+}$ bath condition, the Peredox LT was *decreased* when $Ca^{2+}_{CYT}$ was elevated but *increased* following external $Na^+$ re-application (*Figure 1E*), which suggests that glycolysis activation is strongly tied to the activity of the $Na^+/K^+$ pump, but not strongly tied to the activity of $Ca^{2+}$ pumps (nor to the increase in $Ca^{2+}_{CYT}$ per se, e.g., via a signaling mechanism).

Our ion substitution experiments also hinted that both glycolysis activation and $Ca^{2+}_{CYT}$ regulation in DGCs are dependent on forward NCX transport. A $Ca^{2+}_{CYT}$ increase (from reverse NCX transport) returned faster to baseline levels if external $Na^+$ was present (i.e., if forward NCX was activated, *Figure 1F*; and compare *Figure 1C* with *Figure 1—figure supplement 1A*), and $NADH_{CYT}$ did not increase following a $Ca^{2+}_{CYT}$ increase if external $Ca^{2+}$ was removed without re-applying external $Na^+$ (*Figure 1—figure supplement 1A and B*). $NADH_{CYT}$ production increased when external $Na^+$ was removed and re-applied without previously elevating $Ca^{2+}_{CYT}$ (i.e., without activating forward NCX, *Figure 1—figure supplement 1C and D*), but this $NADH_{CYT}$ increase was weaker than when external $Na^+$ was re-applied while $Ca^{2+}_{CYT}$ was elevated (i.e., when forward NCX was activated, *Figure 1D and E*).

The inability of a $Ca^{2+}_{CYT}$ elevation to measurably stimulate $NADH_{CYT}$ production in the absence of external $Na^+$ suggests that any increase in activity of PMCA and SERCA to pump $Ca^{2+}$ from the cytoplasm does not substantially activate glycolysis, even during the prolonged $Ca^{2+}_{CYT}$ increases in our ion substitution experiments. But maybe inhibiting one of these $Ca^{2+}$ pumps, to divert more $Ca^{2+}_{CYT}$ to be extruded by the non-inhibited pump, could boost the activity of the other and consequently amplify its associated effect on glycolysis. We tested this possibility by inhibiting SERCA, which compared to PMCA has more specific pharmacology and uses less ATP per $Ca^{2+}$ transported. Thapsigargin (1 µM, a specific SERCA inhibitor) increased both the $NADH_{CYT}$ and $Ca^{2+}_{CYT}$ transients evoked by antidromic electrical stimulations in ACSF (*Figure 1—figure supplement 2A and B*) as well as the ΔPeredox/ΔR-CaMP ratio (*Figure 1—figure supplement 2C*). These effects are consistent with SERCA inhibition and a diversion of $Ca^{2+}_{CYT}$ to other clearance pathways that are either more closely coupled to glycolysis or that transport fewer ions per ATP utilized. However, reverse-NCX-mediated $Ca^{2+}_{CYT}$ increases in the presence of 1 µM thapsigargin (and absence of external $Na^+$) still failed to increase $NADH_{CYT}$ (*Figure 1—figure supplement 1B*), meaning that any putative amplification of PMCA activity after SERCA inhibition was still not enough to drive measurable glycolysis activation.

We have previously shown that electrical stimulation increases DGC mitochondrial $Ca^{2+}$ ($Ca^{2+}_{MITO}$) from $Ca^{2+}$ entry through the mitochondrial $Ca^{2+}$ uniporter (MCU) (*Díaz-García et al., 2021a*). Could $Ca^{2+}_{CYT}$'s failure to activate $NADH_{CYT}$ production during our ion substitution experiments be due to a lack of activation of some unknown $Ca^{2+}_{MITO}$-dependent process? This seems unlikely, as bath-applied 0.5 mM external $Ca^{2+}$ in the absence of external $Na^+$ increased the fluorescence LT of mitochondrially targeted RCaMP (*Figure 1—figure supplement 3A and B*), indicating that $Ca^{2+}_{MITO}$ is increased by reverse-NCX-mediated $Ca^{2+}$ entry. $Ca^{2+}_{MITO}$ slightly decreased when external $Ca^{2+}$ was removed, but complete return of $Ca^{2+}_{MITO}$ to baseline levels was not achieved until external $Na^+$ was applied, demonstrating that $Ca^{2+}_{MITO}$ regulation is dependent on $Na^+$, like $Ca^{2+}_{CYT}$.

In summary, these ion substitution experiments show that neuronal $NADH_{CYT}$ production is closely coupled to the flux of $Na^+$, and that a $Ca^{2+}_{CYT}$ elevation is not sufficient to elevate $NADH_{CYT}$ regardless of whether it is pumped by SERCA, pumped by PMCA, or taken up into mitochondria via MCU. They also show that $NADH_{CYT}$ production is activated more strongly by conditions that favor forward

NCX transport that exchanges $Ca^{2+}_{CYT}$ for $Na^+$, and that regulation of both DGC $Ca^{2+}_{CYT}$ and $Ca^{2+}_{MITO}$ depends on $Na^+$. These observations suggest that neuronal glycolysis is both strongly coupled to the activity of the $Na^+/K^+$ pump and sensitive to $Na^+$ import via forward NCX transport, which would become activated following an excitatory event where $Ca^{2+}_{CYT}$ must be removed.

### Malate-aspartate shuttle activity does not mask a $Ca^{2+}$-induced increase to $NADH_{CYT}$

The malate-aspartate shuttle (MAS) is a major pathway that recycles $NADH_{CYT}$ by transferring reducing equivalents from $NADH_{CYT}$ to mitochondria (*McKenna et al., 2006*; *Satrústegui and Bak, 2015*). MAS activity is activated by $Ca^{2+}_{CYT}$ elevations but is reduced by $Ca^{2+}_{MITO}$ elevations (*Contreras and Satrústegui, 2009*; *Satrústegui and Bak, 2015*), which likely means that neuronal excitation would cause a brief MAS activation followed by a longer inhibition, mirroring the $Ca^{2+}_{CYT}$ and $Ca^{2+}_{MITO}$ transient time courses (*Contreras and Satrústegui, 2009*). But the state of MAS activity in our ion substitution experiments was uncertain since reverse NCX transport produced prolonged increases to both $Ca^{2+}_{CYT}$ and $Ca^{2+}_{MITO}$ (*Figure 1* and *Figure 1—figure supplement 3*), so it remained a possibility that

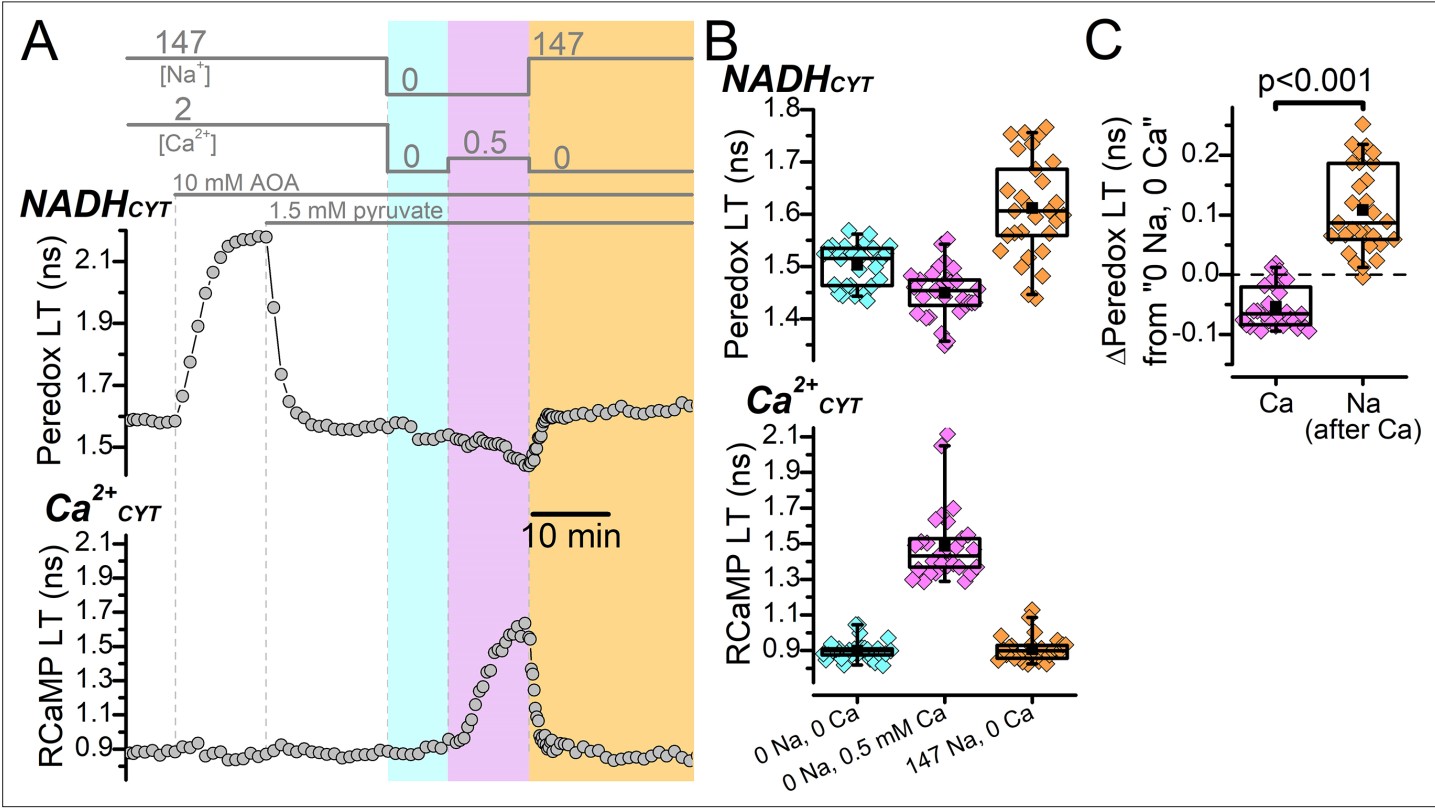

**Figure 2.** Malate-aspartate shuttle activity does not conceal a possible $NADH_{CYT}$ increase produced by elevated $Ca^{2+}_{CYT}$. (**A**) Representative fluorescence lifetime (LT) traces of Peredox (*top*) and RCaMP (*bottom*) from a DGC. External solution changes are indicated by the bars above the Peredox trace. The recording begins in ACSF (147 mM $Na^+$ and 2 mM $Ca^{2+}$), where the addition of 10 mM AOA strongly increases the Peredox LT. Addition of 1.5 mM pyruvate reverted the Peredox LT approximately back to baseline. Following this, external $Na^+$ and $Ca^{2+}$ were removed (*cyan shading*), which slightly decreased the Peredox LT. The addition of 0.5 mM $Ca^{2+}$ in 0 $Na^+$ to stimulate reverse NCX transport increased the RCaMP LT and further decreased the Peredox LT (*magenta shading*). Subsequently, the removal of external $Ca^{2+}$ and re-addition of external $Na^+$ to stimulate forward NCX transport (*orange shading*) increased the Peredox LT and brought the RCaMP LT back to baseline. (**B**) Box plots of the Peredox (*top*) and RCaMP (*bottom*) fluorescence LTs in the different external solution conditions performed in (**A**), across many DGCs (n=29). The external [$Na^+$] and [$Ca^{2+}$] condition after applying AOA and pyruvate is listed across the bottom in chronological order from left to right. The coloring of the data corresponds to the shading in A. The mean Peredox and RCaMP LTs in each external condition were (in ns): 1.50±0.04 in 0 $Na^+$ and 0 $Ca^{2+}$, 1.45±0.05 in 0 $Na^+$ and 0.5 mM $Ca^{2+}$, and 1.61±0.09 in 147 $Na^+$ and 0 $Ca^{2+}$. The mean RCaMP LTs were (in ns): 0.90±0.06 in 0 $Na^+$ and 0 $Ca^{2+}$, 1.49±0.20 in 0 $Na^+$ and 0.5 mM $Ca^{2+}$, and 0.91±0.07 in 147 $Na^+$ and 0 $Ca^{2+}$. (**C**) Relative change to the Peredox fluorescence LT induced by the addition of 0.5 mM $Ca^{2+}$ in the absence of external $Na^+$ to stimulate reverse NCX transport (Ca) or 147 mM $Na^+$ in the absence of external $Ca^{2+}$ to stimulate forward NCX transport (Na). The data are from the same DGCs as in (**B**). The mean ΔPeredox values were: –0.05±0.04 ns for 'Ca' and 0.11±0.07 ns for 'Na'. Statistical significance from a paired Wilcoxon test is indicated.

$Ca^{2+}_{CYT}$ activation of MAS could increase $NADH_{CYT}$ recycling and conceal any small increase to $NADH_{CYT}$ production due to increased activity of $Ca^{2+}$ pumps. Therefore, we tested if a $Ca^{2+}_{CYT}$ elevation could increase $NADH_{CYT}$ after MAS inhibition (*Figure 2*).

Representative fluorescence LT traces from a DGC show that Peredox LT was substantially increased by the application of AOA to block the shuttle's aspartate aminotransferase enzyme (*Figure 2A*), as previously described, *Díaz-García et al., 2017*, which is consistent with an inhibition of $NADH_{CYT}$ recycling and consequent elevation of the cytoplasmic $NADH:NAD^+$. This strong increase to Peredox LT, which approaches the upper end of its dynamic range, precludes the ability to measure further $NADH_{CYT}$ increases, so we applied exogenous pyruvate (1.5 mM) to promote the oxidation of NADH to $NAD^+$ via lactate dehydrogenase and restore the Peredox LT approximately back to its baseline LT in ACSF. We then removed external $Na^+$ and $Ca^{2+}$, which decreased the Peredox LT from the AOA and pyruvate condition (*Figure 2A*); and then increased $Ca^{2+}_{CYT}$ by applying 0.5 mM external $Ca^{2+}$ in zero external $Na^+$ to drive reverse NCX transport. We confirmed the $Ca^{2+}_{CYT}$ increase from the increase to RCaMP LT, but the $Ca^{2+}_{CYT}$ elevation still did not coincide with an increase to $NADH_{CYT}$. Rather, $NADH_{CYT}$ decreased relative to the 0 $Na^+$ 0 $Ca^{2+}$ baseline (*Figure 2B and C*). This demonstrates that an increase to $Ca^{2+}_{CYT}$ still fails to observably stimulate $NADH_{CYT}$ production even when $NADH_{CYT}$ recycling by the MAS is attenuated. Following the $Ca^{2+}_{CYT}$ increase, we removed 0.5 mM external $Ca^{2+}$ and re-applied external $Na^+$ to activate forward NCX transport (*Figure 2A*), which increased the Peredox LT (*Figure 2B and C*). This shows again that $NADH_{CYT}$ production is coupled to the flux of $Na^+$.

Overall, these experiments argue against a concealment of $Ca^{2+}_{CYT}$-induced $NADH_{CYT}$ production by a $Ca^{2+}_{CYT}$ activation of $NADH_{CYT}$ recycling via the MAS and indicates, once again, that an increase to $Na^+/K^+$ pump activity is the major driver of glycolysis.

## Activation of glycolysis by $Ca^{2+}$ is mediated by coupling between the $Na^+/Ca^{2+}$-exchanger and the $Na^+/K^+$ pump

The increase to $NADH_{CYT}$ production by an influx of $Na^+$ (*Figure 1* and *Figure 2*) suggests that glycolysis activation in DGCs is strongly tied to an increase in the activity of the $Na^+/K^+$ pump. Also, the $NADH_{CYT}$ increase induced by external $Na^+$ application was stronger if $Ca^{2+}_{CYT}$ was elevated compared to when $Ca^{2+}_{CYT}$ was *not* elevated (*Figure 1E*), meaning that forward NCX transport, which trades $Ca^{2+}_{CYT}$ for $Na^+$, can potentiate glycolysis activation. By considering these observations together with the apparent insensitivity of glycolysis to increases in $Ca^{2+}$ pump activity, it seemed logical that a transient $Ca^{2+}_{CYT}$ elevation would activate glycolysis predominantly by stimulating $Na^+$ influx through forward NCX transport, which would then increase the activity of the $Na^+/K^+$ pump.

We sought for a way to directly test how $NADH_{CYT}$ is affected by inhibition of the $Na^+/K^+$ pump or inhibition of forward NCX transport (*Figure 3*). $Na^+/K^+$ pumps are specifically inhibited by ouabain, although the rodent $Na^+/K^+$ pump α1-isozymes have relatively low sensitivity and require millimolar inhibitor concentrations for complete inhibition under physiological conditions (*Blanco and Mercer, 1998*; *Marks and Seeds, 1978*). Application of 2 mM ouabain to DGCs bathed in ACSF quickly led to $Ca^{2+}$ overload and cell rupture (*Figure 3—figure supplement 1*), which made it impossible to measure how electrical stimulation-induced $NADH_{CYT}$ transients are affected by complete $Na^+/K^+$ pump inhibition. Clearly, the activity of the $Na^+/K^+$ pump must be substantial, even at rest, but achieving complete $Na^+/K^+$ pump inhibition under physiological neuronal conditions is challenging. DGCs bathed in ACSF were more tolerant to application of low-dose (5 μM) ouabain, which inhibits only the rodent $Na^+/K^+$ pump α2- and α3-isozymes that have higher inhibitor sensitivities (*Blanco and Mercer, 1998*), but the stimulation-induced $NADH_{CYT}$ transient was not eliminated (*Figure 3—figure supplement 2*). This result argues that in these cells, the α1 subunit is adequate to support the activation of glycolysis.

We successfully established a condition amenable to millimolar ouabain application by reducing the external $[Na^+]$ to 40 mM and removing external $Ca^{2+}$ to prevent unintended reverse NCX transport. Stimulation-induced $Ca^{2+}_{CYT}$ transients using an electrode were not possible for DGCs bathed in this condition, but we evoked similar $Ca^{2+}_{CYT}$ transients by applying puffs of extracellular $Ca^{2+}$ to transiently produce reverse-NCX transport (*Figure 3A*). A brief puff of $Ca^{2+}$ to a DGC within a slice bathed in 40 mM $Na^+$ and 0 $Ca^{2+}$ (*Figure 3B*) induced a fast transient increase to the RCaMP LT, reflecting a fast increase to $Ca^{2+}_{CYT}$, followed by a transient increase to the Peredox LT, reflecting an increase to $NADH_{CYT}$. This shows that a transient $Ca^{2+}_{CYT}$ elevation can, indeed, activate glycolysis under these ionic conditions. But for a DGC within a slice that was instead bathed in zero external $Na^+$ to prevent

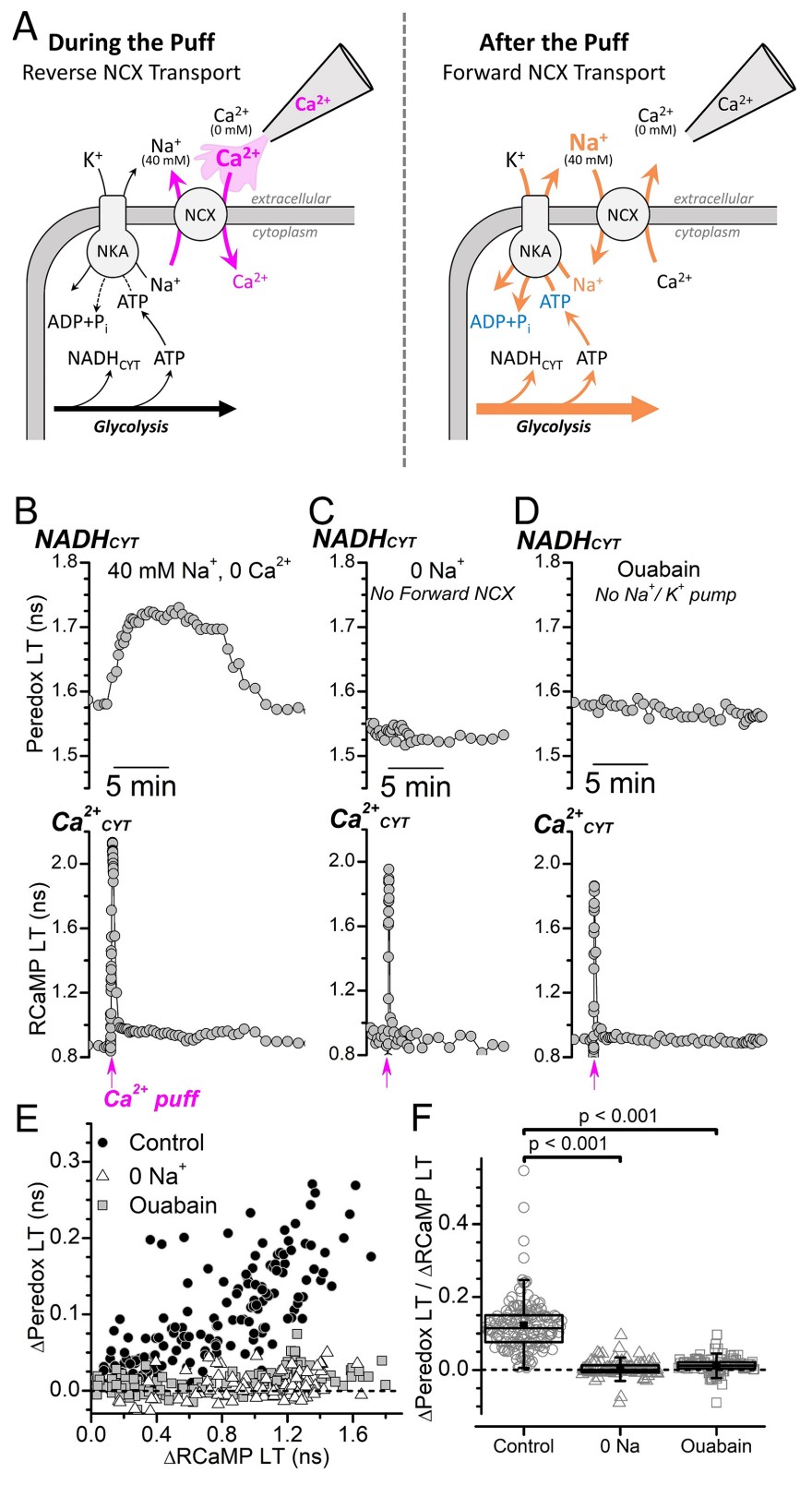

**Figure 3.** Ca$^{2+}_{CYT}$ transients induced by external Ca$^{2+}$ puffs only increase NADH$_{CYT}$ production when both forward NCX transport and Na$^+$/K$^+$ pump activity are intact. (**A**) Cartoon depicting the expected effect of a local external Ca$^{2+}$ puff on ion transport by the NCX and subsequent activation of the Na$^+$/K$^+$ pump. (*Left*) A pipette containing Ca$^{2+}$ is placed near the soma of a DGC within a slice that is bathed in solution containing 40 mM Na$^+$ and 0 Ca$^{2+}$

*Figure 3 continued on next page*

*Figure 3 continued*

solution. A brief pulse of positive pressure is applied to the pipette to transiently increase the local external [$Ca^{2+}$] and stimulate reverse NCX transport (*magenta arrows*), leading to $Ca^{2+}$ import and an increase to $Ca^{2+}_{CYT}$. (*Right*) In the aftermath of the puff, the local external [$Ca^{2+}$] decreases as the small volume of puffed $Ca^{2+}$ mixes with large volume of the 40 mM $Na^+$ 0 $Ca^{2+}$ bath solution, which leads to forward NCX transport that stimulates $Na^+$ extrusion and an increase to $Na^+/K^+$ pump activity (*orange arrows*). Abbreviations: $Na^+/Ca^{2+}$-exchanger (NCX), $Na^+/K^+$-ATPase (NKA). (**B–D**) Representative fluorescence lifetime (LT) traces of Peredox (*top traces*) and RCaMP (*bottom traces*) from a DGC bathed in either 40 mM $Na^+$ and 0 $Ca^{2+}$ solution (**B**, *Control*), 0 $Na^+$ and 0 $Ca^{2+}$ to block forward NCX transport (**C**, *0 Na+*), or in 40 mM $Na^+$ and 0 $Ca^{2+}$ with 5 mM ouabain to block the $Na^+/K^+$ pump (**D**, *Ouabain*). Puffs of $Ca^{2+}$ were delivered at the timepoint indicated along the bottom of the RCaMP LT traces (*magenta arrows*). (**E**) Scatterplot of the $Ca^{2+}$-puff-induced transient changes to the Peredox and RCaMP LTs of DGCs in 40 mM $Na^+$ and 0 $Ca^{2+}$ (*Control, black circles,* n=172), in 0 $Na^+$ and 0 $Ca^{2+}$ (*0 Na+, white triangles*, n=102), or in 40 mM $Na^+$ and 0 $Ca^{2+}$ with 5 mM ouabain (*Ouabain, gray squares*, n=114). The dashed line indicates where ΔPeredox=0. Analysis was restricted to recordings with transient RCaMP LT half-decay times <35 s. (**F**) Ratio of ΔPeredox to ΔRCaMP for the same $Ca^{2+}$-puff-induced LT transients as in panel E, except excluding those recordings where the ΔRCaMP LT was <0.2 ns to avoid noisy ratio calculations due to small ΔRCaMP values. The dashed line indicates where ΔPeredox/ΔRCaMP=0. The mean ΔPeredox/ΔRCaMP in each condition was: 0.12±0.07 in 40 mM $Na^+$ and 0 $Ca^{2+}$ (*Control*, n=147), 0.01±0.02 in 0 $Na^+$ and 0 $Ca^{2+}$ (*0 Na+*, n=96), and 0.00±0.02 in 40 mM $Na^+$ and 0 $Ca^{2+}$ with 5 mM ouabain (*Ouabain*, n=99). Statistical significance from Mann-Whitney tests is indicated.

The online version of this article includes the following figure supplement(s) for figure 3:

**Figure supplement 1.** Complete $Na^+/K^+$ pump inhibition in ACSF leads to cell rupture.

**Figure supplement 2.** Effect of low dose (5 μM) ouabain on transient Peredox and RCaMP lifetime (LT) increases evoked by synaptic or antidromic stimulation.

the NCX-mediated influx of $Na^+$ in exchange for the elevated $Ca^{2+}_{CYT}$, a $Ca^{2+}$ puff still evoked a fast $Ca^{2+}_{CYT}$ transient but did not increase $NADH_{CYT}$ (*Figure 3C*). Similarly, the $NADH_{CYT}$ increase that would normally follow the $Ca^{2+}$ puff was prevented for a DGC within a slice bathed with 5 mM ouabain to block $Na^+/K^+$ pumps (*Figure 3D*).

A scatter plot of the transient Peredox LT and RCaMP LT changes that were evoked by $Ca^{2+}$ puffs (*Figure 3E*) illustrates the positive correlation between $NADH_{CYT}$ production and $Ca^{2+}_{CYT}$ increases for DGCs bathed in 40 mM $Na^+$ and 0 $Ca^{2+}$, where both forward NCX and $Na^+/K^+$ pump activities are intact. But $NADH_{CYT}$ production was deeply attenuated when DGCs were bathed without external $Na^+$ or with 5 mM ouabain, even with large elevations of $Ca^{2+}_{CYT}$ (ΔRCaMP>1.2 ns). The mean ΔPeredox/ΔRCaMP ratio (*Figure 3F*) for $Ca^{2+}$-puff-evoked transients in 40 mM external $Na^+$ (0.122±0.005, n=147) was reduced to nearly 0 after blocking forward NCX transport by removing external $Na^+$ (0.003±0.002, n=96) or after blocking the $Na^+/K^+$ pump with ouabain (0.013±0.002, n=99).

These data demonstrate that $Ca^{2+}_{CYT}$ can activate glycolysis in the DGC soma if it is traded for $Na^+$ through forward NCX transport, which increases the activity of the $Na^+/K^+$ pump. The lack of $Ca^{2+}$-puff-induced $NADH_{CYT}$ responses when either forward NCX transport or the $Na^+/K^+$ pump was blocked further substantiates that any increase in the activity of $Ca^{2+}$ pumps has a negligible effect on glycolysis activation.

## Antidromic stimulation-induced $Na^+$ transients depend on $Ca^{2+}$

The stimulation of $NADH_{CYT}$ production by $Ca^{2+}_{CYT}$ through coupling between forward NCX transport and the $Na^+/K^+$ pump (*Figure 3*) indicates that $Na^+$ influx during forward NCX transport can activate glycolysis. But how does the amount of $Na^+$ imported by the NCX to extrude $Ca^{2+}_{CYT}$ compare to the total $Na^+$ influx induced by neuronal excitation? We investigated this by measuring intracellular [$Na^+$] changes during electrical stimulation with and without $Ca^{2+}$ entry (*Figure 4A*).

We recorded stimulation-induced changes in intracellular [$Na^+$] and $Ca^{2+}_{CYT}$ simultaneously by loading SBFI (*Harootunian et al., 1989*), a $Na^+$-sensitive fluorescence dye, into RCaMP-expressing DGCs via single-cell electroporation. Representative traces of the SBFI relative fluorescence intensity (ΔF/F) and RCaMP LT from a DGC bathed in ACSF show that antidromic stimulation evoked a transient decrease to SBFI ΔF/F (note the inverted y-axis), reflecting an increase to [$Na^+$], and a transient increase to RCaMP LT, reflecting an increase to $Ca^{2+}_{CYT}$ (*Figure 4B*). SBFI ΔF/F transients were longer in duration than the RCaMP LT transient, regardless of whether they were evoked by antidromic stimulation or synaptic stimulation in the absence of synaptic blockers (*Figure 4* and *Figure 4—figure*

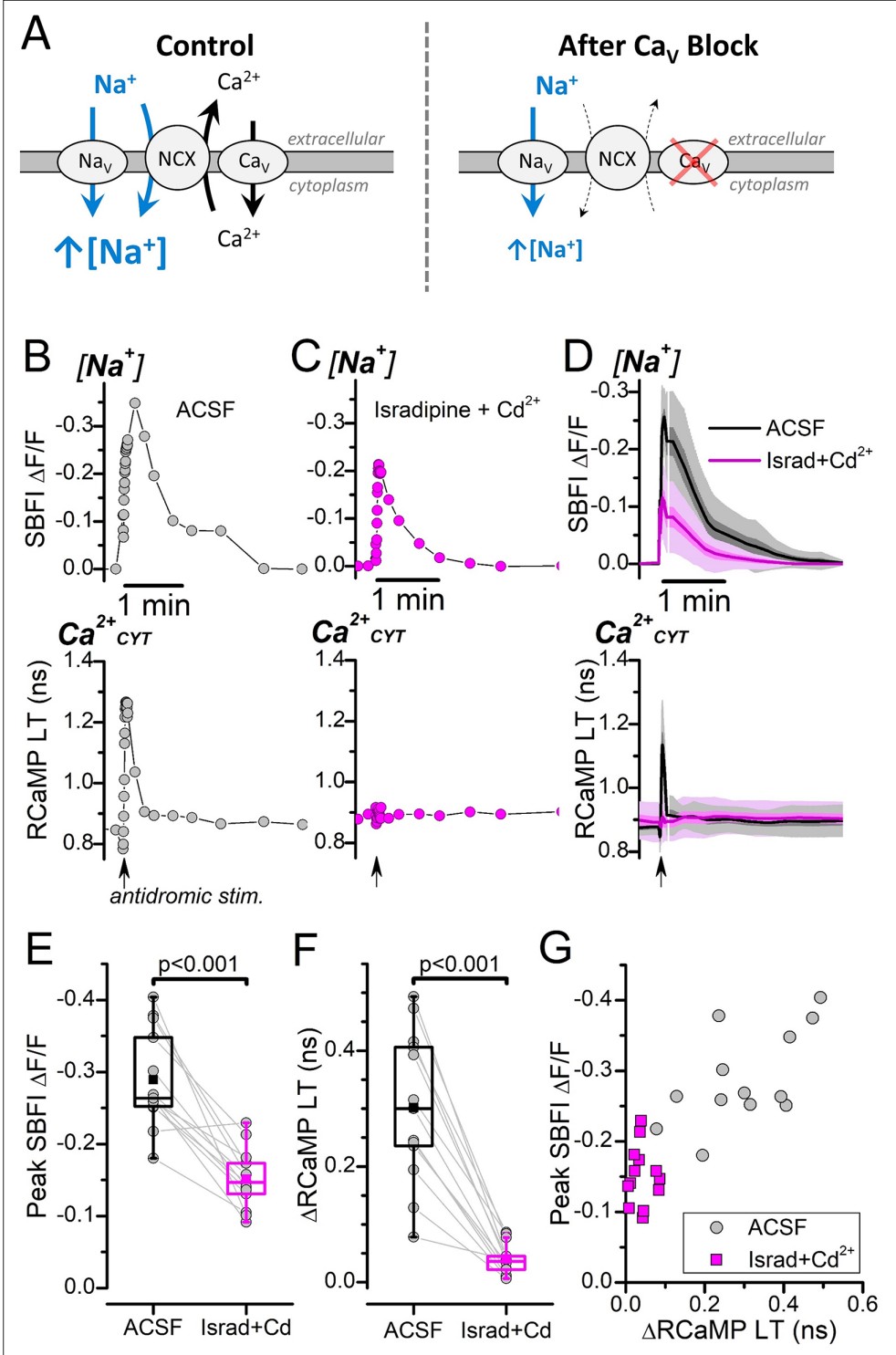

**Figure 4.** Antidromic stimulation-induced [Na+] transients depend strongly on Ca2+ entry. (**A**) Cartoon depicting antidromic stimulation-induced Na+ and Ca2+ fluxes. *Left, Control*: Stimulation in ACSF triggers Na+ and Ca2+ influx through their respective voltage-gated Na+ and Ca2+ channels. Following the Ca2+CYT elevation, the NCX will import Na+ to drive Ca2+ extrusion. Both the Na+ influx through NaV and through the NCX contribute to the total intracellular [Na+] increase (*blue arrows*). *Right, After CaV Block*: Inhibiting Ca2+ influx through CaV (indicated by the *red X*) prevents Ca2+CYT elevation, which strongly reduces Na+ influx through the NCX. Abbreviations: voltage-gated Na+ channel (NaV), voltage-gated Ca2+ channel (CaV), Na+/Ca2+-exchanger (NCX). (**B**) Representative fluorescence traces showing antidromic stimulation-induced transient changes to SBFI ΔF/F (*top*) and RCaMP lifetime (LT)

*Figure 4 continued on next page*

*Figure 4 continued*

(*bottom*) from a DGC bathed in ACSF with synaptic blockers. The stimulation was delivered at the timepoint indicated by the arrow at the bottom of the RCaMP trace. The SBFI $\Delta F/F$ axis is inverted to illustrate an increase to $[Na^+]$ as an upward deflection. (**C**) Antidromic stimulation-induced transient changes to SBFI $\Delta F/F$ and RCaMP LT in the presence of 3 µM isradipine and 20 µM $CdCl_2$ for the same DGC as in (**B**). The stimulation was delivered at the timepoint indicated by the arrow below the RCaMP trace. (**D**) Average, interpolated, antidromic stimulation-induced SBFI $\Delta F/F$ and RCaMP LT transients (n=13) before (*ACSF, black trace and shading*) and after $Ca_V$ block (*Israd+Cd$^{2+}$, magenta trace and shading*). The means are indicated by the solid line, SEM is indicated by the darker shading, and SD is indicated by the lighter shading. (**E**) Paired box plots showing the antidromic stimulation-induced peak transient SBFI $\Delta F/F$ amplitudes before (*ACSF, black box*) and after $Ca_V$ block (*Israd+Cd$^{2+}$, magenta box*) for the same DGCs as in (**D**). The mean peak SBFI $\Delta F/F$ values were: $-0.29\pm0.07$ in ACSF and $-0.15\pm0.04$ with Israd+Cd$^{2+}$. Statistical significance from a paired sample t-test is indicated. (**F**) Paired box plots showing the $\Delta$RCaMP LT before (*ACSF, black box*) after $Ca_V$ block (*Israd+Cd$^{2+}$, magenta box*) for the same DGCs as in (**D**) and (**E**). The mean $\Delta$RCaMP LTs were: $0.30\pm0.13$ ns in ACSF and $0.04\pm0.03$ ns with Israd+Cd$^{2+}$. Statistical significance from a paired sample t-test is indicated. (**G**) Scatterplot of antidromic stimulation-induced transient changes to SBFI $\Delta F/F$ against $\Delta$RCaMP LT before (*ACSF, gray circles*) and after $Ca_V$ block (*Israd +Cd$^{2+}$, magenta squares*). Data are from the same DGCs as in (**D–F**).

The online version of this article includes the following figure supplement(s) for figure 4:

**Figure supplement 1.** Somatic intracellular $[Na^+]$ and $Ca^{2+}_{CYT}$ transients induced by synaptic or antidromic stimulation.

---

*supplement 1*), indicating that stimulation-induced intracellular $[Na^+]$ elevation outlasts $Ca^{2+}_{CYT}$ elevation. A scatterplot of the transient changes to SBFI $\Delta F/F$ and RCaMP LT shows that intracellular $[Na^+]$ and $Ca^{2+}_{CYT}$ changes are positively correlated when evoked by either stimulation paradigm (***Figure 4— figure supplement 1D***); while this is consistent with a positive dependence of intracellular $[Na^+]$ on $Ca^{2+}_{CYT}$, a positive correlation between $[Na^+]$ and $Ca^{2+}_{CYT}$ increases would also be observed if neuronal stimulation evokes proportional yet independent changes to both ions.

To directly test if somatic intracellular $[Na^+]$ changes depend on $Ca^{2+}$ entry, we measured how antidromic stimulation-induced $[Na^+]$ transients were affected by blocking $Ca^{2+}$ channels (***Figure 4A***). Blocking voltage-gated $Ca^{2+}$ channels with isradipine and $Cd^{2+}$ (***Berjukow et al., 2000***; ***Lansman et al., 1986***) attenuated the antidromic stimulation-induced RCaMP LT transient (***Figure 4C and D***), indicating a strong reduction to $Ca^{2+}$ entry and $Ca^{2+}_{CYT}$ elevation. The SBFI $\Delta F/F$ transient became smaller after blocking $Ca^{2+}$ entry (***Figure 4C and D***), demonstrating that stimulation-induced intracellular $[Na^+]$ changes depend on $Ca^{2+}_{CYT}$ elevation; the peak of the SBFI $Na^+$ transient (as $\Delta F/F$) was reduced to $54.6\pm19.3\%$ of the original response and the $Ca^{2+}$ transient (as $\Delta$RCaMP LT) was reduced to $14.5\pm12.5\%$ (n=13) (***Figure 4E and F***).

A scatterplot of the antidromic stimulation-induced transient changes to SBFI $\Delta F/F$ and $\Delta$RCaMP LT in ACSF or in the presence of isradipine and $Cd^{2+}$ (***Figure 4G***) shows that blocking $Ca^{2+}$ entry shifts the peak SBFI $\Delta F/F$ down and to the left, which indicates that a major component of somatic $[Na^+]$ changes is dependent on $Ca^{2+}_{CYT}$. This means that $Na^+$ imported by forward NCX transport in response to $Ca^{2+}_{CYT}$ extrusion comprises a substantial proportion of the $Na^+$ that enters the soma following stimulation, which provides clarity to the apparent dependence of antidromic stimulation-induced glycolysis activation on $Ca^{2+}_{CYT}$ elevation (***Díaz-García et al., 2021a***): an elevation to $Ca^{2+}_{CYT}$ potentiates $Na^+$ influx, which ultimately leads to a stronger activation of the $Na^+/K^+$ pump than when $Ca^{2+}$ entry is blocked.

## Discussion

Glycolysis in neurons is transiently increased following stimulation (***Díaz-García et al., 2017***; ***Fox et al., 1988***; ***Fox and Raichle, 1986***). To understand the mechanism that drives this increase to glycolysis following neuronal electrical activity, we tested the responsiveness of $NADH_{CYT}$ to changes in intracellular $Na^+$ and $Ca^{2+}_{CYT}$, which affect energy consumption by ion-ATPases. We have shown in hippocampal DGC somas that $NADH_{CYT}$ is strongly influenced by changes in intracellular $Na^+$ and that activation of $NADH_{CYT}$ increases by $Ca^{2+}_{CYT}$ is nearly completely dependent on ion transport coupling between the NCX and the $Na^+/K^+$ pump. This means that nearly all of the transient glycolysis increase

following a stimulation is a response to an activation of the Na$^+$/K$^+$ pump to extrude the Na$^+$ that enters either through Na$^+$ channels or through the NCX to power active Ca$^{2+}$ transport.

## Neuronal glycolysis is strongly influenced by the Na$^+$/K$^+$ pump, but not by Ca$^{2+}$ pumps

Coupling between glycolysis and Na$^+$/K$^+$ pump activity has been reported across many tissues and cell types (*Andersen and Marmarou, 1992*; *Balaban and Bader, 1984*; *Hasin and Barry, 1984*; *Hellstrand et al., 1984*; *James et al., 1996*; *James et al., 1999*; *Knull, 1978*; *Lipton and Robacker, 1983*; *Lynch and Balaban, 1987*; *Mercer and Dunham, 1981*; *Minakami et al., 1964*; *Paul et al., 1979*; *Paul, 1983*; *Proverbio and Hoffman, 1977*; *Weiss and Hiltbrand, 1985*; *Whittam et al., 1964*; *Whittam and Ager, 1965*). Our lab previously reported a link between the activity of the Na$^+$/K$^+$ pump α3-isozyme and glycolysis activation in DGCs of acute mouse hippocampal slices: the potentiation to antidromic stimulation-induced NADH$_{CYT}$ production after increasing channel-mediated influx of Na$^+$ with α-pompilidotoxin in the absence of external Ca$^{2+}$ (i.e., when both Na$^+$ influx through the NCX and Ca$^{2+}$ influx through channels are blocked) could be reversed by strophanthidin at low micromolar concentrations (*Díaz-García et al., 2021a*). We have shown here in DGCs that NADH$_{CYT}$ production decreases with depletion of Na$^+$ and increases with repletion of Na$^+$ (*Figure 1*), which strongly suggests that the rate of neuronal glycolysis mirrors changes in intracellular [Na$^+$] and in the activity of the Na$^+$/K$^+$ pump. Moreover, DGCs could not withstand complete Na$^+$/K$^+$ pump inhibition when bathed in ACSF (*Figure 3—figure supplement 1*), meaning that Na$^+$/K$^+$ pump activity is essential to counteract Na$^+$ leak and/or influx through Na$^+$-coupled transporters even when neurons are at rest.

The small NADH:NAD$^+$ increase produced by removing only external Ca$^{2+}$ that we observed during our previous studies (*Díaz-García et al., 2021a*) is also consistent with a predominant influence of Na$^+$ on glycolysis: since Na$^+$ conductance through the NALCN (Na$^+$-leak channel, non-selective), which regulates DGC resting membrane potential and excitability (S.-Y. *Lee et al., 2019*), is higher in the absence of external Ca$^{2+}$ (*Kschonsak et al., 2020*; *Lu et al., 2010*), removing external Ca$^{2+}$ likely elevates Na$^+$ influx through the NALCN and increases Na$^+$/K$^+$ pump activity.

The sensitivity of glycolysis to Ca$^{2+}$ pump activity is less clear. The small increases to the antidromic stimulation-induced ΔPeredox/ΔRCaMP ratio after applying inhibitors that target SERCA (thapsigargin; *Figure 1—figure supplement 2*) or PMCA (E6-berbamine or calmidazolium; *Brini et al., 2013*; *Díaz-García et al., 2021a*) do suggest some involvement of Ca$^{2+}$ pumps in regulating stimulation-induced Ca$^{2+}_{CYT}$ increases, but our observations that NADH$_{CYT}$ was not increased by the elevation of Ca$^{2+}_{CYT}$ alone, either by reverse NCX transport during ion substitution or Ca$^{2+}$ puffs in either zero external Na$^+$ or 5 mM ouabain, indicate that ATP consumption associated with the direct pumping of Ca$^{2+}$ does not have a measurable effect on glycolysis (*Figures 1–3*, and *Figure 1—figure supplement 1*). This may indicate that Ca$^{2+}$ pumps are either not closely coupled to glycolytic enzymes in DGCs or that Ca$^{2+}$ pump density is small with respect to Na$^+$/K$^+$ pumps. It appears that the density of Na$^+$/K$^+$ pumps outweighs the density of Ca$^{2+}$ pumps in hippocampus since hippocampal homogenates have ouabain-sensitive ATPase activity ~2- to 3-fold larger than Ca$^{2+}$-sensitive ATPase activity (*Gutierres et al., 2014*; *Spohr et al., 2022*; *Teixeira et al., 2020*). Even if we assume that Ca$^{2+}$ pumps and Na$^+$/K$^+$ pumps are coupled to glycolysis equally, the lower density of Ca$^{2+}$ pumps would make their glycolytic (and total cellular) ATP consumption less than the ATP consumed by Na$^+$/K$^+$ pumps.

Ca$^{2+}$ pumps have an apparent $K_{0.5}$ for Ca$^{2+}$ activation of ~0.1 μM and relatively slow maximal turnover rates of ~100 s$^{-1}$ (characteristic of ion pumps; *Glitsch, 2001*; *Meyer et al., 2017*), meaning that they are well suited for tuning the resting Ca$^{2+}_{CYT}$ but not for regulating large dynamic Ca$^{2+}_{CYT}$ increases (*Caroni and Carafoli, 1981*). It is likely that Ca$^{2+}$ pump transport capacity would saturate quickly in response to the 0.3–1 μM increases in Ca$^{2+}$ during some stimulations. If an increase to Ca$^{2+}$ pump activity was the predominant driver of NADH$_{CYT}$ production, the saturation of their Ca$^{2+}$ transport during larger Ca$^{2+}_{CYT}$ increases would likely cause a 'rounding off' of the Peredox LT change for stimulations where RCaMP LT changes are >0.5 ns; but the near linearity of the ΔPeredox-ΔRCaMP relationship for RCaMP LT changes of 0.05–1 ns (*Díaz-García et al., 2017*) argues against such a scenario.

# The $Ca^{2+}_{CYT}$-dependence of aerobic glycolysis activation arises from ion transport coupling between the $Na^+/Ca^{2+}$-exchanger and the $Na^+/K^+$ pump

Our observations here that glycolysis activation in DGCs is strongly coupled to the activity of the $Na^+/K^+$ pump but not $Ca^{2+}$ pumps was seemingly at odds with the ~70% reduction to antidromic stimulation-induced $NADH_{CYT}$ production by blocking $Ca^{2+}_{CYT}$ elevation (*Díaz-García et al., 2021a*). However, the reduction to $NADH_{CYT}$ production is fully explained by the involvement of forward NCX transport in DGC $Ca^{2+}_{CYT}$ regulation (*Figure 1F*; *Lee et al., 2012*) and by the close coupling between the ion transport of the NCX and the $Na^+/K^+$ pump.

The NCX has an apparent $K_{0.5}$ for internal $Ca^{2+}$ of ~1 µM (*Blaustein and Santiago, 1977*; *Collins et al., 1992*; *Miura and Kimura, 1989*) and a transport rate 10–30 times faster than the slow turnover rates of ATPases (*Blaustein and Lederer, 1999*; *Caroni et al., 1980*). This means that NCX transport activity would be near the lower end of its total capacity when neurons are at rest (when $Ca^{2+}_{CYT}$ is ~100 nM) but could rapidly respond to stimulation-induced $Ca^{2+}_{CYT}$ increases and exchange it for $Na^+$. Unfortunately, most NCX inhibitors are unspecific or do not block the forward transport mode (*Iwamoto et al., 2004*; *Iwamoto and Kita, 2006*; *Secondo et al., 2015*; *Sharikabad et al., 1997*), so the most effective way to test the involvement of forward NCX transport in glycolysis activation is to replace external $Na^+$ with an inert ion that cannot be transported by the NCX (i.e., choline).

The transient $NADH_{CYT}$ production evoked by $Ca^{2+}$ puffs in 40 mM $Na^+$ and 0 $Ca^{2+}$ was eliminated when external $Na^+$ was completely removed (i.e., replaced with choline) or when ouabain was added (*Figure 3*), meaning that both forward NCX transport and $Na^+/K^+$ pump activity are required for a $Ca^{2+}_{CYT}$ elevation to measurably activate glycolysis. This unequivocally demonstrates that the exchange of $Ca^{2+}_{CYT}$ for $Na^+$ by the NCX is closely coupled to $Na^+$ extrusion by the $Na^+/K^+$ pump, and that this coupling is tightly associated with the activation of glycolysis. Coupling between the NCX and $Na^+/K^+$ pump within discrete microdomains has also been reported in cardiac tissue (*Mohler et al., 2005*; *Zhang et al., 2009*; *Zhang et al., 2011*).

Strong coupling between the NCX, the $Na^+/K^+$ pump, and glycolysis activation can also explain two of our other observations: First, it can explain why the stimulation-induced ΔPeredox/ΔRCaMP ratio is increased by inhibition of SERCA or PMCA. Blocking either of these $Ca^{2+}$ pumps likely increases the $Ca^{2+}$ that is exported by the NCX, which would amplify the associated $Na^+$ influx through the NCX and accelerate the ATP consumption by the $Na^+/K^+$ pump to extrude $Na^+$. Second, it can explain why $NADH_{CYT}$ production was decreased when $Ca^{2+}_{CYT}$ was elevated by reverse NCX transport (*Figure 1* and *Figure 1—figure supplement 1*). For reverse NCX transport to import 1 $Ca^{2+}$ from the extracellular environment, the NCX must export 3 $Na^+$ (*Figure 1A*). This means that an increase to $Ca^{2+}_{CYT}$ would occur at the same time as a decrease to intracellular $[Na^+]$, which would decrease the activity of the $Na^+/K^+$ pump and, consequently, decrease the rate of glycolysis and its associated $NADH_{CYT}$ production. The decrease to cytosolic $NADH:NAD^+$ from reverse NCX transport further supports the strong association between the energy consumed to extrude $Na^+$ and glycolysis and implies that even decreasing the $Na^+/K^+$ pump activity from an already reduced steady-state level of activity (due to the depletion of intracellular $[Na^+]$ from prior application of the 0 Na 0 Ca solution) is more influential on the rate of glycolysis than any putative strong activation of $Ca^{2+}$ pumps by a $Ca^{2+}_{CYT}$ increase.

The ΔPeredox/ΔRCaMP ratio evoked by $Ca^{2+}$ puffs in 40 mM $Na^+$ and 0 $Ca^{2+}$ (~0.1, *Figure 3E*) is ~40–50% of the ΔPeredox/ΔRCaMP ratio evoked by electrical stimulation in ACSF (~0.2–0.25) (*Díaz-García et al., 2017*; *Díaz-García et al., 2021a*), meaning that the peak activation of glycolysis relative to the $Ca^{2+}_{CYT}$ change is smaller when evoked by $Ca^{2+}$ puffs compared to electrical stimulation. This could be due to the absence of channel-mediated $Na^+$ entry during $Ca^{2+}$ puffs compared to electrical stimulation or to the reduced external $[Na^+]$ in $Ca^{2+}$ puff experiments (40 mM) compared to the external $[Na^+]$ during electrical stimulation in ACSF (147 mM), which would slow down the kinetics of $Ca^{2+}_{CYT}$-dependent $Na^+$ influx through forward NCX transport, consequently leading to a less potent activation of the $Na^+/K^+$ pump; both the quantity and the rate of $Na^+$ influx are likely to be important determinants of the $NADH_{CYT}$ transient amplitude and time course.

Glycolysis does not appear to be influenced by intracellular $[Na^+]$ per se, but rather by the downstream effect of $[Na^+]$ on $Na^+/K^+$ pump activity. If glycolytic enzymes could be directly stimulated by an increase to $[Na^+]$, then complete inhibition of the $Na^+/K^+$ pump should have facilitated stronger $NADH_{CYT}$ responses. But this was not the case, as $NADH_{CYT}$ responses evoked by $Ca^{2+}$ puffs were

strongly attenuated in the presence of 5 mM ouabain (*Figure 3*), a condition that allows for Na⁺ influx through the NCX but prevents Na⁺ extrusion through the Na⁺/K⁺ pump. In contrast to the strong attenuation by 5 mM ouabain, the increase to the NADH$_{CYT}$ transients evoked by synaptic electrical stimulation after applying 5 µM ouabain in ACSF (*Figure 3—figure supplement 2*) is not as straightforward and cannot be attributed solely to an inhibition of the Na⁺/K⁺ pump. First, 5 µM ouabain does not substantially inhibit rodent α1-pumps (*Marks and Seeds, 1978*; *Sweadner, 1979*); we know that 5 µM ouabain only causes partial Na⁺/K⁺ pump inhibition in our acute slice preparation from the mouse because it does not cause Ca²⁺$_{CYT}$ overload (unlike 5 mM ouabain in ACSF), which means that the plasma membrane Na⁺ gradient in the presence of 5 µM ouabain remains sufficient to prevent a reversal of NCX transport. Second, the stimulation-induced Ca²⁺$_{CYT}$ transients were increased, which is, perhaps, reflective of an effect on synaptic release.

We think that the absence of strong NADH$_{CYT}$ responses to Ca²⁺$_{CYT}$ increases when Na⁺ is unavailable for exchange via NCX (*Figure 1* and *Figure 3*) is quite informative on how both ions influence glycolysis activation, even though it is true that the rapid ionic changes that occur through channels during neuronal activity following electrical stimulation might obey somewhat different rules than the slower ionic changes we are able to make by driving ion fluxes through a secondary ion transporter. Since we have determined that NADH$_{CYT}$ production is strongly increased by an influx of Na⁺ but not increased by large increases in Ca²⁺$_{CYT}$ (*Figures 1–3*) it is logical to infer that a transient increase to NADH$_{CYT}$ production following electrical stimulation is primarily coupled to the energy consumed to directly pump Na⁺ rather than to directly pump Ca²⁺. The amplitudes of RCaMP LT increases that we achieve with both the ion substitution experiments (*Figure 1*) and the Ca²⁺ puff experiments (*Figure 3*) are also similar to the typical amplitude of RCaMP LT increase when evoked by electrical stimulation (*Figure 4* and *Díaz-García et al., 2017*; *Díaz-García et al., 2021a*), meaning that Ca²⁺$_{CYT}$ is increased to similar levels by all experimental paradigms regardless of their site of entry (i.e., channels or NCX).

## Na⁺ entry via the Na⁺/Ca²⁺-exchanger contributes substantially to total Na⁺ entry in the soma

It seemed that stimulation-induced Na⁺ influx via the NCX could be substantial given that a transient Ca²⁺$_{CYT}$ increase could only activate glycolysis if both forward NCX transport and the Na⁺/K⁺ pump were active (*Figure 3*) and that blocking Ca²⁺$_{CYT}$ strongly reduces the NADH$_{CYT}$ response (*Díaz-García et al., 2021a*). The ~50% reduction to the stimulation-induced SBFI ΔF/F peak amplitudes after blocking Ca²⁺ entry confirmed that Na⁺ influx via forward NCX transport contributes greatly to the total Na⁺ influx in the soma (*Figure 4*). We think it is unlikely that the reduction to the [Na⁺] transient after blocking Ca²⁺ entry resulted from less Na⁺ entering through voltage-gated Ca²⁺ channels, since Na⁺ permeation through these channels is blocked by external Ca²⁺ with a K$_d$ of ~1 µM (*Almers and McCleskey, 1984*; *Tsien et al., 1987*).

It is interesting to note that the positive relationship between the amplitudes of stimulation-induced transient increases to NADH$_{CYT}$ and Ca²⁺$_{CYT}$ appears to intercept the origin (cf. *Figure 2*; *Díaz-García et al., 2017*) while the relationship between the amplitudes of transient intracellular [Na⁺] and Ca²⁺$_{CYT}$ increases does not (*Figure 4—figure supplement 1*). Perhaps weaker stimulations that evoke small increases to intracellular [Na⁺] without a measurable increase to Ca²⁺$_{CYT}$ have only very weak effects on glycolysis: it could be that smaller [Na⁺] increases have proportionally more clearance by diffusion through the cytoplasm (*Hodgkin and Keynes, 1956*; *Pusch and Neher, 1988*) or that the energy needed to regulate smaller [Na⁺] increases (without sustained Na⁺ influx through forward NCX transport) is mostly derived from ATP buffering systems known to compartmentalize near sites of high ATP turnover (*Kültz and Somero, 1995*; *Lange et al., 2002*; *Ovádi and Saks, 2004*; *Wallimann et al., 1992*), which could subsequently replenish their energy storage pools without large steady-state changes to glycolytic flux. Directly evaluating the role of these ATP buffering enzymes, that is, creatine kinase and adenylate kinase, in shaping the stimulation-induced transient glycolysis activation in neurons will be a key area for future exploration.

## Estimation of neuronal stimulation-induced changes to intracellular [Na⁺] and Na⁺/K⁺ pump activity

SBFI calibration curves in rodent hippocampal neurons (*Baeza-Lehnert et al., 2019*; *Diarra et al., 2001*; *Gerkau et al., 2019*; *Meier et al., 2006*; *Rose et al., 1999*) reported apparent Na⁺ K$_d$ values

of 18–42 mM Na$^+$. However, most were performed with ouabain concentrations of only 50–100 μM, well below the millimolar concentrations needed to fully inhibit the rodent Na$^+$/K$^+$ pump α1 isozyme under physiological conditions; the relatively high ouabain-resistance of rodent α1 pumps would also be strongly increased in calibration solutions that substitute external Na$^+$ with high concentrations of K$^+$ since external K$^+$ competes with binding of the inhibitor to the pump's externally accessible E2 conformation (*Hansen and Skou, 1973*; *Kanai et al., 2021*).

Despite the limitations of these in situ SBFI calibrations in rodent neurons, we use Rose et al.'s calibration of SBFI ΔF/F excited at 790 nm in brain slices with a Na$^+$ $K_d$ of 26 mM (*Rose et al., 1999*) to *estimate* the stimulation-induced [Na$^+$] changes in our experiments (which reflect the net movement of Na$^+$ through influx and extrusion pathways). Our estimations assume a baseline intracellular [Na$^+$] of 13 mM, which is the reported value for hippocampal slice CA1 neurons (*Mondragão et al., 2016*) and nearly identical to the $K_{0.5}$ for activation of internal Na$^+$-dependent, ATP-induced current produced by α1β1 Na$^+$/K$^+$ pumps (which likely set baseline neuronal internal [Na$^+$]; *Azarias et al., 2013*; *Blanco, 2005*; *Blanco and Mercer, 1998*) in excised patches when internal Na$^+$ is substituted with K$^+$ (*Meyer et al., 2017*; *Meyer et al., 2019*; *Meyer et al., 2020*).

The mean peak amplitude of antidromic stimulation-induced SBFI ΔF/F transients evoked in ACSF was –25.6±4.6% (*Figure 4*), reflecting an average transient [Na$^+$] increase of ~25 mM from baseline. How would this intracellular [Na$^+$] change affect neuronal Na$^+$/K$^+$ pump activity? Neurons express the α1 and α3 isoforms of the Na$^+$/K$^+$ pump (*Blanco and Mercer, 1998*). Under physiological conditions, both α1 and α3 pumps are rate limited by the binding of internal Na$^+$ since their $K_{0.5,Na}^+$ is similar to, or above, physiological intracellular [Na$^+$] while their $K_{0.5,K}^+$ is below physiological extracellular [K$^+$]. Activation of Na$^+$/K$^+$ pumps by internal Na$^+$ is also highly cooperative (Hill coefficient of ~3, corresponding to three transported Na$^+$ *Crambert et al., 2000*; *Hasler et al., 1998*; *Meyer et al., 2017*; *Meyer et al., 2019*; *Meyer et al., 2020*). Based on the reported curves of Na$^+$/K$^+$ pump activation by internal Na$^+$ (*Crambert et al., 2000*; *Meyer et al., 2019*), a 25 mM increase to neuronal [Na$^+$] from a baseline of 13 mM would likely increase peak α1β1 activity ~2-fold (to ~95% of total activity) and α3 activity ~9-fold (to ~60% of total activity).

Blocking Ca$^{2+}$ entry reduced the antidromic stimulation-induced peak SBFI ΔF/F to –11.5±5.5% (*Figure 4*), a ~55% decrease from the transient in ACSF that corresponds nonlinearly to a reduction of the average [Na$^+$] increase from ~25 to ~8 mM. The estimated 8 mM stimulation-induced increase to internal [Na$^+$] when Ca$^{2+}$ entry is inhibited (i.e., when Na$^+$ influx via forward NCX transport is reduced) would increase α1β1 activity ~1.5-fold (to ~80% of total activity) and α3 activity ~3-fold (to ~25% of total activity).

It must be noted that the proportional influence of NCX-mediated Na$^+$ entry on glycolysis activation could be dependent on the stimulation paradigm. Antidromic stimulation in the presence of synaptic blockers does not involve entry of Na$^+$ through NMDA or AMPA receptors. This means that most channel-mediated Na$^+$ influx that occurs from this stimulation method is through voltage-gated channels, which, in DGCs, are highly localized to the axon initial segment with weak, if any, staining in the soma (*Kole et al., 2008*; *Kress et al., 2008*; *Kress et al., 2010*). Thus, it is not unexpected that Na$^+$ influx via forward NCX transport has such a prominent effect on the antidromic stimulation-induced glycolysis activation in the soma. Soma SBFI transients evoked by either antidromic or synaptic stimulation could reach similar peak amplitudes (*Figure 4—figure supplement 1*), but the contribution of Na$^+$ influx through forward NCX transport to synaptically evoked [Na$^+$] changes could not be tested because blocking voltage-gated Ca$^{2+}$ entry would prevent synaptic vesicle release.

NCX involvement in glycolysis activation could also vary in different neuronal compartments. Reported SBFI transient measurements indicate that antidromic stimulation-induced [Na$^+$] changes in the soma are typically smaller than transients in the dendrites (*Fleidervish et al., 2010*; *Kole et al., 2008*). Measuring the ion and metabolite dynamics in dendrites will be an important step for understanding whether the mechanism of glycolysis activation in dendrites is similar to or different than the mechanism in somas.

## On the mechanism of neuronal stimulation-induced aerobic glycolysis activation in somas

Based on our observations here and previously (*Díaz-García et al., 2017*; *Díaz-García et al., 2021a*) regarding neuronal stimulation-induced [Na$^+$], Ca$^{2+}$$_{CYT}$, and NADH$_{CYT}$ dynamics in the soma, we can

begin to compile a more comprehensive understanding of the mechanism underlying transient aerobic glycolysis activation in response to neuronal excitation.

Neuronal stimulation immediately increases both intracellular $[Na^+]$ and $Ca^{2+}_{CYT}$. The transient $Ca^{2+}_{CYT}$ elevation lasts at most a few seconds (slightly longer than the duration of the stimulus; RCaMP1h has a $t_{1/2}$ decay of 410 ms; *Akerboom et al., 2013*) before returning to baseline due to buffering by cytosolic proteins and removal from the cytosol. This means that any transient activation of $Ca^{2+}$ pumps by $Ca^{2+}_{CYT}$ would likely only be a few seconds in duration before returning to steady state. In any case, we know that this produces no measurable activation of glycolysis since similar $Ca^{2+}_{CYT}$ transients evoked by $Ca^{2+}$ puffs in the absence of external $Na^+$ or presence of ouabain were not associated with $NADH_{CYT}$ increases. It is also clear that $Ca^{2+}_{CYT}$ is rapidly exported across the plasma membrane by forward NCX transport, which loads the neuron with $Na^+$. $Na^+$ is not buffered in the cytoplasm and is actively extruded back across the plasma membrane exclusively by the $Na^+/K^+$ pump. The transient intracellular $[Na^+]$ increase lasts for a minute or more (longer than the $Ca^{2+}_{CYT}$ transient), during which an increase to the activity of $Na^+/K^+$ pumps would be sustained. Both the sensitivity of $NADH_{CYT}$ production to $Na^+$ and the ablation of $Ca^{2+}_{CYT}$-evoked $NADH_{CYT}$ transients by inhibition of either forward NCX transport or the $Na^+/K^+$ pump demonstrate that pumping $Na^+$ is the predominant activity that drives the transient activation of aerobic glycolysis.

Stimulation also triggers a rapid increase to $Ca^{2+}_{MITO}$ via MCU that can last several minutes (*Díaz-García et al., 2021a*) and requires $Na^+$ in order to return to the baseline (*Figure 1—figure supplement 3*). The requirement of $Na^+$ for $Ca^{2+}_{MITO}$ extrusion likely indicates a $Na^+$-coupled $Ca^{2+}$ efflux mechanism, as reported in other cell types (*Palty and Sekler, 2012*; *Parpura et al., 2016*; *Saotome et al., 2005*). Interestingly, the attenuation of $Ca^{2+}_{MITO}$ elevation from MCU knockdown reduces the stimulation-induced cytosolic ΔPeredox/ΔRCaMP by ~50% (*Díaz-García et al., 2021a*), which is consistent with $Ca^{2+}_{MITO}$ extrusion stimulating glycolysis. The extent to which the energetic burden of $Ca^{2+}_{MITO}$ extrusion is placed onto glycolysis, and, more specifically, the $Na^+/K^+$ pump, or elsewhere will be an important area for future exploration.

Given the close coupling of the $Na^+/K^+$ pump to neuronal glycolysis, it will be exciting to measure how changes in $Na^+/K^+$ pump activity affect the flux of other glycolytic metabolites using biosensors that report lactate (*Koveal et al., 2022*; *San Martín et al., 2013*), pyruvate (*San Martín et al., 2014*), and glucose (*Díaz-García et al., 2019*), or how changes in $Na^+/K^+$ pump activity affect cellular energy status (i.e., [ATP] [*Imamura et al., 2009*] and ATP:ADP [*Tantama et al., 2013*]). These investigations will ultimately provide a more complete picture of the relationship between ion homeostasis and metabolism.

# Materials and methods

### Key resources table

| Reagent type (species) or resource | Designation | Source or reference | Identifiers | Additional information |
|---|---|---|---|---|
| Strain, strain background (*Mus musculus*, M and F) | C57BL/6NCrl | Charles River | RRID:IMSR_CRL:27 | |
| Recombinant DNA reagent | AAV.CAG.Peredox.WPRE.SV40 | *Mongeon et al., 2016* | | Addgene #73807 |
| Recombinant DNA reagent | AAV.hSyn.RCaMP1h.WPRE.SV40 | *Akerboom et al., 2013* | | |
| Recombinant DNA reagent | AAV.hSyn.mito-RCaMP1h.WPRE.SV40 | *Díaz-García et al., 2021a* | | |
| Chemical compound, drug | Isradipine | Abcam | Cat: ab120142, CAS: 75695-93-1 | |
| Chemical compound, drug | NBQX (6-Nitro-7- sulfamoylbenzo [f]quinoxaline-2,3-dione, Disodium Salt) | Toronto Research Chemicals | Cat: N550005, CAS: 479347-86-9 | |
| Chemical compound, drug | D-AP5 (D-(-)–2-Amino-5-phosphonopentanoic acid) | Abcam | Cat: ab120003, CAS: 79055-68-8 | |
| Chemical compound, drug | Picrotoxin | Sigma-Aldrich | Cat: P1675 CAS: 124-87-8 | |

*Continued on next page*

*Continued*

| Reagent type (species) or resource | Designation | Source or reference | Identifiers | Additional information |
|---|---|---|---|---|
| Chemical compound, drug | CdCl₂ | Sigma-Aldrich | Cat: 202908, CAS: 10108-64-2 | |
| Chemical compound, drug | Poly-L-lysine | Sigma-Aldrich | Cat: P4832 | |
| Chemical compound, drug | Aminooxyacetate (O-(carboxymethyl) hydroxylamine hemihydrate) | Sigma-Aldrich | Cat: C13408 CAS: 2921-14-4 | |
| Chemical compound, drug | Pyruvic acid | Sigma-Aldrich | Cat: 107360, CAS: 127-17-3 | |
| Chemical compound, drug | Thapsigargin | Santa Cruz Biotechnology | Cat: sc-24017A, CAS: 67526-95-8 | |
| Chemical compound, drug | Ouabain | Sigma-Aldrich | Cat: O3125, CAS: 11018-89-6 | |
| Chemical compound, drug | SBFI K⁺ salt (fluorescent dye) | Ion Biosciences | Cat: 2022B | |
| Other | Glass capillaries, Borosilicate, standard wall, no filament, 4 in., O.D. 1.5 mm | WPI | Cat: 1B150-4 | For microelectrodes and pipettes |
| Other | Glass coverslips, 12 mm circle No.1 | VWR | Cat: 48366–251 | For brain slice handling |

## Animals

Experiments were performed using male and female wild-type mice (C57BL/6NCrl, Charles River Laboratories), which were housed in a barrier facility in individually ventilated cages with ad libitum access to standard chow (PicoLab 5053). All experiments followed approved IACUC protocols and the NIH Guide for the Care and Use of Laboratory Animals and Animal Welfare Act. All procedures were approved by the Harvard Medical Area Standing Committee on Animals.

## Viral vectors

DNA constructs encoding Peredox, RCaMP, or mito-RCaMP biosensors were packaged into adeno-associated virus (AAV) vectors using either the Penn Vector Core at University of Pennsylvania or the Viral Core Facility at Children's Hospital in Boston, MA. AAV vectors encoding RCaMP were also produced in our laboratory, as previously described (*Kimura et al., 2019*). The AAV8 serotype was used for Peredox and AAV9 for RCaMP and mito-RCaMP. AAVs were aliquoted and stored at –80°C.

## Intracranial injections

Postnatal day 1 or 2, mice were injected intracranially with AAV to express biosensors in the hippocampus (*Díaz-García et al., 2021b*). Mice were cryoanesthetized and injected with 150 nL AAV using an UltraMicroPump III (WPI, Sarasota, FL) microinjector in two locations of each hemisphere at the following coordinates relative to lambda: (1) 0 mm anterior-posterior; ±1.9 mm medial-lateral; –2.0 mm dorsal-ventral; and (2) 0 mm anterior-posterior; ±2.0 mm medial-lateral; –2.3 mm dorsal-ventral. Injected pups were placed on a heating pad and allowed to recover before being returned to their cages and were administered daily ketoprofen (10 mg/kg) subcutaneously for up to 3 days.

## Hippocampal slice preparation

Injected mice between 14 and 24 days old were anesthetized with isoflurane, decapitated, and the brain was removed into ice-cold slicing solution containing (in mM) 87 NaCl, 2.5 KCl, 1.25 NaH₂PO₄, 25 NaHCO₃, 75 sucrose, 25 D-glucose, 0.5 CaCl₂, and 7 MgCl₂ (~335 mOsm/kg) and bubbled with 95% O₂ and 5% CO₂. The brain was glued by the dorsal side, embedded into 2% agarose in phosphate-buffered saline, and submerged in a chamber with ice-cold slicing solution. Horizontal 275 µm brain slices were cut using a compresstome (VF-310-0Z, Precisionary) and immediately transferred to a chamber with ACSF containing (in mM) 120 NaCl, 2.5 KCl, 1 NaH₂PO₄, 26 NaHCO₃, 10 D-glucose, 2 CaCl₂, and 1 MgCl₂ (~290 mOsm/kg) that was warmed to 36°C and bubbled with 95% O₂ and 5% CO₂. The slices rested on a mesh bottom to adequately perfuse both sides of the tissue. After 35 min, the chamber was cooled to room temperature and the brain slices therein were used for the next 4 hr.

## Recording solutions and pharmacology

The brain slices were adhered to glass coverslips coated with poly-L-lysine (P4832, Sigma-Aldrich), placed in a bath chamber mounted to the microscope, and superfused with solutions maintained at 33–34°C and bubbled with 95% $O_2$ and 5% $CO_2$ at a rate of 5 mL/min. All solutions were ~290 mOsm/kg and contained 25 μM D-AP5, 5 μM NBQX, and 100 μM picrotoxin (unless otherwise specified) to block synaptic transmission. $Ca^{2+}$-free solution contained (in mM) 120 NaCl, 2.5 KCl, 1 $NaH_2PO_4$, 26 $NaHCO_3$, 10 D-glucose, and 4.1 $MgCl_2$. $Na^+$- and $Ca^{2+}$-free solution contained (in mM) 120 Choline-Cl, 1.5 KCl, 1 $KH_2PO_4$, 26 Choline-$HCO_3$, 10 D-glucose, and 4.1 $MgCl_2$. 1 mM EGTA was added to the $Ca^{2+}$-free solutions from a 0.5 M stock.

For $Ca^{2+}$-puff experiments, the 40 mM $Na^+$ and $Ca^{2+}$-free solution was obtained by mixing $Ca^{2+}$-free solution with $Na^+$- and $Ca^{2+}$-free solution; all solutions for those experiments also contained 0.1 mM EGTA.

Thapsigargin (1 μM) was added from a 10 mM stock in DMSO. Aminooxyacetate (AOA, 10 mM) was added from a 2 M stock in water. Pyruvate (1.5 mM) was added from a 1 M aqueous stock, pH 7.3 with N-methyl-D-glucamine. Ouabain was either directly dissolved (for 2–5 mM) prior to recording or added from a 5 mM stock in water (for 5 μM). Isradipine (3 μM) was added from a 50 mM DMSO stock. $CdCl_2$ (20 μM) was added from a 100 mM stock in water.

## Electrical stimulation

Stimulation trains (of 100 μs pulse width) were delivered as previously described (*Díaz-García et al., 2017*; *Díaz-García et al., 2021a*) using a concentric bipolar electrode (FHC, Bowdoin, ME) mounted on a motorized micromanipulator (Burleigh PCS-6000, ThorLabs, Sterling, VA) and connected to an A360 stimulus isolation unit (WPI, Sarasota, FL). DGCs were stimulated antidromically (100 pulses, 50 Hz, 750–1500 μA) by placing the electrode in the hippocampal hilus, or synaptically (60 pulses, 20 Hz, 100–500 μA) by placing the electrode in the molecular layer.

## $Ca^{2+}$ puffs

Borosilicate glass pipettes (with a tip diameter of 5 μm and resistance of 1 MΩ when filled with ACSF) were fabricated on a micropipette puller (P-97 Flaming/Brown, Sutter Instruments), backfilled with 100–250 mM $CaCl_2$ and placed into a pipette holder mounted on a motorized micromanipulator with a closed pressure system connected to a 1 mL syringe. The pipette tip was placed adjacent to groups of DGCs expressing Peredox and RCaMP biosensors. Pressure delivery was controlled by a solenoid valve (NResearch) in series with the syringe line that was triggered by a digital/analog actuator (NTE Electronics). Positive pressure was applied using the syringe, and $Ca^{2+}$ puffs were delivered by opening the solenoid for 0.5–5 s beginning 300 ms after the start of image acquisition.

## Single-cell electroporation of SBFI

Sodium-binding benzofuran isophthalate (SBFI) is a fluorescent $Na^+$-indicator (*Harootunian et al., 1989*). We loaded the SBFI $K^+$ salt (Ion Biosciences, San Marcos, TX) version into DGCs using single-cell electroporation methods derived from *Nevian and Helmchen, 2007*; we chose this method rather than loading by patch pipette to avoid diluting cytoplasmic metabolites and soluble expressed biosensor, and also to avoid potentially toxic effects on the slice from applying Pluronic-F127 and DMSO to load the membrane-permeant SBFI-acetoxymethyl ester version.

Borosilicate glass pipettes were fabricated (with a tip diameter of 1 μm and resistance of 3 MΩ when filled with ACSF), loaded with 4 mM SBFI-$K^+$ in distilled $H_2O$, and placed onto a pipette holder with a closed pressure system mounted on a motorized micromanipulator. The SBFI solution was contacted by a chlorided silver wire connected to an A360 stimulus isolation unit, and an Ag-AgCl reference electrode was placed in the bath chamber and connected to the same stimulation device. The pipette tip was placed directly next to the soma of an RCaMP-expressing DGC. No positive pressure was applied. A single 10 ms pulse of negative 1.0–1.2 μA loaded the DGC with SBFI. Recovery of the membrane potential from these electroporation pulses takes ~1–2 min (*Nevian and Helmchen, 2007*). We waited ≥15 min after the electroporation to start the image acquisition and electrically stimulate the SBFI-loaded, RCaMP-expressing DGCs.

The intracellular [dye] from electroporation is estimated to be ~20% of the pipette concentration (*Nevian and Helmchen, 2007*). Thus, the intracellular [$Na^+$] will not be substantially buffered by

the ~0.5–1 mM [SBFI] in our recordings (*Fleidervish et al., 2010*; *Mondragão et al., 2016*), meaning that intracellular [Na$^+$] is set by cellular mechanisms and SBFI merely responds to changes in [Na$^+$]. Also, the complexation of Na$^+$ with crown ether molecules (the Na$^+$-sensitive moiety of SBFI) is nearly diffusion-controlled, with very rapid rates of formation and dissociation (*Adamic et al., 1986*; *Liese-gang et al., 1977*). This means that the response time of SBFI fluorescence changes should be fast compared to almost all neuronal processes.

## Two-photon fluorescence imaging

Fluorescence imaging data were acquired using a Thorlabs Bergamo II microscope (Thorlabs Imaging Systems, Sterling, VA) equipped with an Olympus LUMPLFLN 60×/W (NA 1.0) objective lens, hybrid photodetectors R11322U-40 (Hamamatsu Photonics, Shizuoka, Japan), and a Chameleon Vision-S tunable Ti-Sapphire mode-locked laser (80 MHz, ~75 fs pulses; Coherent, Santa Clara, CA). The excitation wavelength was 790 nm. Fluorescence emission light from the Peredox and RCaMP biosensors was split with an FF562-Di03 dichroic mirror and bandpass filtered for green (FF01-525/50) and red (FF01-641/75) light; a red 670/50 bandpass filter was used for experiments where SBFI and RCaMP were multiplexed. The photodetector and laser sync signals were preamplified and then digitized at 1.25 GHz using a field-programmable gate array board (PC720 with FMC125 and FMC122 modules, 4DSP, Austin, TX). A modified version of the ScanImage software written in Matlab (*Pologruto et al., 2003*) (provided by B. Sabatini and modified by G.Y.) controlled the laser, microscope, and image acquisition (128×128 pixels, scanning rate of 2 ms per line).

Time-correlated single-photon counting was performed using laboratory-built firmware and software to determine the arrival time of each photon relative to the laser pulse. The fluorescence LT was determined from a nonlinear least-squares fit to the photon arrival histograms in Matlab (Mathworks, Natick, MA) convolved with a Gaussian for the impulse response function (*Yasuda et al., 2006*). Fluorescence intensity was determined from the total photon counts.

## Data analysis

Fluorescence images were analyzed offline using Matlab R2014b software. Regions of interest (ROIs) were defined around individual DGC somas and photon statistics were calculated for all pixels within the ROI. LT values were calculated by fitting the photon arrival histograms with a biexponential decay function (convolved with a Gaussian for the impulse response function [*Yasuda et al., 2006*] up to 8 ns after the peak photon arrival time). The time constant of this fit is denoted as the 'tau8' value (*Díaz-García et al., 2019*); using tau8 values minimizes both the fit variability (by restricting the averaging to the approximate time window of the actual data) and the variability between the LTs recorded on different experimental setups with different data acquisition windows.

RCaMP signals were unmixed from Peredox as previously described (*Díaz-García et al., 2017*; *Díaz-García et al., 2021a*). RCaMP signals were unmixed from SBFI bleedthrough into the red 670/50 bandpass filter following previous methods (*Díaz-García et al., 2017*) but using an unmixing ratio of 0.053, since the red channel photon counts from SBFI-loaded DGCs (not expressing RCaMP) were (mean ± SD) 5.3±1.2% (n=42) of the green channel photon counts. Stimulation-induced SBFI intensity recordings were analyzed using Origin 8.1 (OriginLab, Northampton, MA) and Python 3 (https://www.anaconda.com/, RRID – SRC:008394, Numpy and Pandas libraries). Individual transient fluorescence intensity traces were baseline subtracted and the peak ΔF/F was determined as the minimum intensity value.

Average time traces of transient fluorescence LT or intensity changes were created using Python. Data acquisition times for individual fluorescence traces were binned to the nearest second for data acquired before and after the stimulation event (which were acquired every 10–60 s) or the nearest 10 ms for data acquired during the stimulation (which were acquired every ~250 ms). Traces from all stimulation-induced transients in the data set were interpolated and merged onto a single time axis, from which the means, standard deviations, and standard error of the means were determined.

Statistical analysis was performed using Origin 8.1 software. Data sets were tested for normality ($\alpha$=0.05) using a Shapiro-Wilk test. Data sets that met normality criteria were compared using a paired t-test, or two-sample t-test. Data sets that did not meet normality criteria were compared using a paired sample Wilcoxon test or Mann-Whitney test for unpaired samples. Data are reported as mean

± standard deviation (unless otherwise indicated). For box plots, the means are indicated by the filled square, medians are indicated by the horizontal bar, and 5–95% ranges are indicated by whiskers.

Figures were created using Origin 8.1 and PowerPoint (Microsoft, Redmond, WA).

## Reagents

All chemical reagents used to make the brain slicing and artificial cerebrospinal fluid (ACSF) solutions were obtained from Sigma-Aldrich (St. Louis, MO).

## Acknowledgements

The authors thank the members of the Yellen lab for their valuable discussions. The authors thank Dr Juan Ramón Martínez François and Dr Pablo Artigas for their critical review of the manuscript. This work was supported by research grants from the National Institutes of Health (R01 NS102586 and R01 GM124038 to GY) and Postdoctoral Fellowships from the National Institutes of Health (F32 NS116105 to DJM and F32 NS100331 to CMDG) and the Harvard Mahoney Neuroscience Institute Fund (to DJM).

## Additional information

### Competing interests

Gary Yellen: Reviewing editor, *eLife*. The other authors declare that no competing interests exist.

### Funding

| Funder | Grant reference number | Author |
| --- | --- | --- |
| National Institute of Neurological Disorders and Stroke | R01 NS102586 | Gary Yellen |
| National Institute of General Medical Sciences | R01 GM124038 | Gary Yellen |
| National Institute of Neurological Disorders and Stroke | F32 NS116105 | Dylan J Meyer |
| National Institute of Neurological Disorders and Stroke | F32 NS100331 | Carlos Manlio Díaz-García |
| Harvard Mahoney Neuroscience Institute | Postdoctoral Fellowship | Dylan J Meyer |

The funders had no role in study design, data collection and interpretation, or the decision to submit the work for publication.

### Author contributions

Dylan J Meyer, Conceptualization, Software, Funding acquisition, Investigation, Methodology, Writing - original draft, Writing – review and editing; Carlos Manlio Díaz-García, Conceptualization, Funding acquisition, Investigation, Methodology, Writing – review and editing; Nidhi Nathwani, Mahia Rahman, Investigation, Methodology, Writing – review and editing; Gary Yellen, Conceptualization, Supervision, Funding acquisition, Methodology, Writing – review and editing

### Author ORCIDs

Dylan J Meyer ⓘ http://orcid.org/0000-0001-8453-3813
Carlos Manlio Díaz-García ⓘ http://orcid.org/0000-0002-4352-2496
Gary Yellen ⓘ http://orcid.org/0000-0003-4228-7866

### Ethics

All experiments were performed in compliance with the NIH Guide for the Care and Use of Laboratory Animals and the Animal Welfare Act. The Harvard Medical Area Standing Committee on Animals

approved all procedures involving animals. (Animal Welfare Assurance Number A3431-01, Protocol IS00001113-3).

## Decision letter and Author response

Decision letter https://doi.org/10.7554/eLife.81645.sa1
Author response https://doi.org/10.7554/eLife.81645.sa2

---

## Additional files

### Supplementary files
• MDAR checklist

### Data availability
All data generated or analysed during this study are included in the manuscript.

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
