## [Editor Report]

The authors investigate mechanisms that regulate glycolytic ATP production in neurons. They conclude that the cytosolic Na^+^, not Ca^2+^, and the activity of the Na^+^/K^+^ pump drive glycolysis. The study is conceptually significant as it seeks to determine how neuronal glycolysis is coupled to electrical activity. The study is thoughtful, uses sophisticated fluorescence lifetime imaging technology and clever experimental designs, and as such provides new insights into how electrical activity regulates glycolysis.

---

## [Decision Letter]

**Decision letter after peer review:**

Thank you for submitting your article "The Na^+^/K^+^ pump dominates control of glycolysis in hippocampal dentate granule cells" for consideration by *eLife*. Your article has been reviewed by 3 peer reviewers, and the evaluation has been overseen by a Reviewing Editor and Richard Aldrich as the Senior Editor. The following individual involved in review of your submission has agreed to reveal their identity: James B Hurley (Reviewer #1).

Essential revisions:

1. It would be helpful to be able to show definitively that the increased level of NADH is not caused by a decrease of NADH consumption (as might occur if metabolism switched from inefficient glycolysis to much more efficient oxidative phosphorylation). I think the experiments evaluating the role of the malate aspartate shuttle do address this, but this specific question (could more efficient consumption of NADH lower the demand and increase steady state NADH) should be addressed in the discussion.

2. It would be helpful to test whether it is just the intracellular concentration of Na^+^ rather than the activity of the Na^+^/K^+^ ATPase that is stimulating glycolysis. The data in Figure 4B-D suggest that the changes in intracellular [Na^+^] are substantial and perhaps they could influence the activity of glycolytic enzymes. The ouabain experiment in Figure 3D does help to show that consumption of ATP by the pump is the driving force. But that set of experiments is complicated by the need for very high concentrations of ouabain that could produce non-specific effects. The authors suggest based on a 1979 paper and a review that oubain cannot fully inhibit the Na^+^/K^+^ ATPases in mouse neurons but they have not shown that directly for the cultured neuron preparation they use in this paper. I think that the interpretations of the ouabain experiments would be more convincing if the authors could show for certain that μM ouabain is not inhibiting the NaK ATPase activity at μM concentrations in the neurons used in this paper. Would it be possible to do that by measuring ATPase activity as was done it the Marks and Seeds 1979 paper that was cited?

3. Please rewrite the last paragraph on p. 19. It seems to be one long sentence and I had to read it several times to decipher the points the authors are making.

4. The NADH/NAD^+^ ratio is used throughout as the only measurement reflecting glycolytic flux.

5. It has been hypothesized that the close association of glycolytic enzymes with ion transporters (such as the Na^+^/K^+^ pump) is meant to provide localized ATP to power these pumps. How does bulk glycolysis (monitored with NADH/NAD^+^ ratio) relate to localized/compartmentalized glycolysis?

6. Related to point 5, most of the peredox measurements in the paper have been made at baseline, in the absence of electrical activity. Therefore, it is not clear how the findings relate to activity-driven glycolysis.

7. The finding that inhibition of SERCA during stimulation actually elevates cytosolic NADH level argues against Na^+^ being the only ion that regulates glycolysis.

8. The finding that "SBFI ΔF/F transients were longer in duration than the RCaMP LT transient" does not necessarily mean that Na^+^ elevation lasts longer than Ca^2+^ in the cell. This could be an artefact of the SBFI on/off rate relative to RCaMP. In fact, prolonged elevation of cytosolic Na^+^ would make neurons refractive to depolarization in AP trains.

9. To ensure that the interpretation of changes in cytosolic NADH are valid with respect to glycolysis, it is important to measure dynamic changes in other metabolites such as glucose.

10. It is not clear why monitoring bulk glycolysis (with NADH/NAD^+^ ratio) is a good method to study localized/compartmentalized glycolysis, which is thought to be closely coupled to the pumps.

11. Related to point 2, most of the peredox measurements in the paper have been made at baseline (albeit under different ionic concentration), in the absence of electrical activity. It is not immediately obvious why electrical activity, which occurs at much faster time scales, would follow the same rules. Indeed SERCA inhibition during activity elevated NADH levels but had no effect at baseline (Figure S1 and S2). How do the authors explain these discrepancies?

12. Pharmacological inhibition of the NCX transporter is needed. Many of the conclusions about its role are based on manipulation of extracellular Ca^2+^/Na^+^ alone.

13. The finding that inhibition of SERCA during stimulation actually elevates cytosolic NADH level argues against Na^+^ being the only ion that regulates glycolysis. This is an interesting finding and there is no need to explain it away. In fact, it may be that it is local microdomains of Ca^2+^ that activate glycolysis (raise NADH level) rather than bulk Ca^2+^.

14. The finding that "SBFI ΔF/F transients were longer in duration than the RCaMP LT transient" does not necessarily mean that Na^+^ elevation lasts longer than Ca^2+^ in the cell. This could be an artefact of the SBFI on/off rate relative to RCaMP. In fact, prolonged elevation of cytosolic Na^+^ would make neurons refractive to depolarization in AP trains.

15. Monensin might be useful to show the effect of a fairly direct activation of the Na^+^/K^+^ pump on glycolytic activity.

16. In addition to altering the ionic compositions, chemical blocking og NCX (SEA0400?) might be useful to validate the role of NCX in the proposed pathway.

17. Switching back from 0 Na^+^ + 0.5 Ca^2+^ to 147 Na^+^ + 0 Ca^2+^ led to a sustained NADHcyto increase. Cytoplasmic [Na^+^] shows only a transient rise when Ca^2+^ entry is inhibited (Figure 4C). Please explain.

18. Please show NADHcyto during antidromic stimulation in presence of Ca_v_ blocker.

19. The cyto-Ca^2+^ response is lower after Ca^2+^ puff in 0Na^+^ and with Ouabain treated condition. What could be the possible explanation?

20. Ca^2+^ activates the glycerol-phosphate shuttle, which also plays role in recycling NADHcyto. I failed to find any experimental data or discussion in the manuscript on how this shuttle could influence the finding.

---

## [Author Response]

Essential revisions:It would be helpful to be able to show definitively that the increased level of NADH is not caused by a decrease of NADH consumption (as might occur if metabolism switched from inefficient glycolysis to much more efficient oxidative phosphorylation). I think the experiments evaluating the role of the malate aspartate shuttle do address this, but this specific question (could more efficient consumption of NADH lower the demand and increase steady state NADH) should be addressed in the discussion.The NADH/NAD^+^ ratio is used throughout as the only measurement reflecting glycolytic flux.Ca^2+^ activates the glycerol-phosphate shuttle, which also plays role in recycling NADHcyto. I failed to find any experimental data or discussion in the manuscript on how this shuttle could influence the finding.

We have previously shown that transient increases to NADH_CYT_ occur regardless of NADH dynamics in mitochondria (Diaz-Garcia et al. 2021 doi:10.7554/*eLife*.64821) and that the NADH_CYT_ responses arise from increased glycolysis rather than a change in the malate-aspartate shuttle (Diaz-Garcia et al. 2017 doi:10.1016/j.cmet.2017.06.021 and results here), so an increase to NADH_CYT_ due to changes in mitochondrial metabolism or decreased NADH consumption seems unlikely. We have now added a paragraph on this to the first section of the Discussion.

Regarding the glycerol-phosphate shuttle: a recent paper using cultured neurons indicates that it does not make a significant contribution to NADH recycling when MCU and/or MAS remain active (Dhoundiyal et al. 2022 doi:10.1523/jneurosci.0193-22.2022), so the G3P shuttle should not have much influence on our results since we never inhibited MCU and MAS simultaneously.

It would be helpful to test whether it is just the intracellular concentration of Na^+^ rather than the activity of the Na^+^/K^+^ ATPase that is stimulating glycolysis. The data in Figure 4B-D suggest that the changes in intracellular [Na^+^] are substantial and perhaps they could influence the activity of glycolytic enzymes. The ouabain experiment in Figure 3D does help to show that consumption of ATP by the pump is the driving force. But that set of experiments is complicated by the need for very high concentrations of ouabain that could produce non-specific effects. The authors suggest based on a 1979 paper and a review that ouabain cannot fully inhibit the Na^+^/K^+^ ATPases in mouse neurons but they have not shown that directly for the cultured neuron preparation they use in this paper. I think that the interpretations of the ouabain experiments would be more convincing if the authors could show for certain that μM ouabain is not inhibiting the NaK ATPase activity at μM concentrations in the neurons used in this paper. Would it be possible to do that by measuring ATPase activity as was done it the Marks and Seeds 1979 paper that was cited?

We do not believe that elevated intracellular Na^+^ alone directly stimulates glycolysis. If this were the case then inhibition of the Na^+^/K^+^ pump (which otherwise limits increases to intracellular Na^+^) during the Ca^2+^ puff experiments should have increased the metabolic transient, but the results show the opposite. Ouabain is a specific inhibitor of the Na^+^/K^+^ pump and applying it at mM concentrations is commonly used to inhibit the more-ouabain-resistant Na^+^/K^+^ pump isozymes (Price and Lingrel 1988 doi:10.1021/bi00422a016, Meyer et al. 2017 doi:10.1085/jgp.201711827, Vedovato and Gadsby 2010 doi:10.1085/jgp.201010407, Azarias et al. 2013 doi: 10.1074/jbc.M112.425785), though we cannot completely rule out possible non-specific effects.

The Marks and Seeds paper reports multiple ouabain binding affinities in freshly obtained brain tissue from mouse, which is a very close match to the acute brain slice tissue (not cultures) used in our experiments.

Please rewrite the last paragraph on p. 19. It seems to be one long sentence and I had to read it several times to decipher the points the authors are making.

We have improved the readability of this paragraph.

It has been hypothesized that the close association of glycolytic enzymes with ion transporters (such as the Na+/K^+^ pump) is meant to provide localized ATP to power these pumps. How does bulk glycolysis (monitored with NADH/NAD^+^ ratio) relate to localized/compartmentalized glycolysis?It is not clear why monitoring bulk glycolysis (with NADH/NAD^+^ ratio) is a good method to study localized/compartmentalized glycolysis, which is thought to be closely coupled to the pumps.

Even if ATP is produced by glycolysis within a submembrane domain, it is likely that the NADH and NAD^+^ associated with this process readily equilibrate throughout the cytosol and nucleus, especially because much of the NADH produced by glycolysis is ultimately reoxidized by mitochondria via the MAS: this requires that NADH and NAD^+^ diffuse from their site of production. The lack of difference in NADH:NAD^+^ between the cytosol and nucleus in our previous report also argues against some forms of compartmentation (cf the images in Figure 1A of Diaz-Garcia et al. 2017 doi:10.1016/j.cmet.2017.06.021).

Related to point 5, most of the peredox measurements in the paper have been made at baseline, in the absence of electrical activity. Therefore, it is not clear how the findings relate to activity-driven glycolysis.Related to point 2, most of the peredox measurements in the paper have been made at baseline (albeit under different ionic concentration), in the absence of electrical activity. It is not immediately obvious why electrical activity, which occurs at much faster time scales, would follow the same rules. Indeed SERCA inhibition during activity elevated NADH levels but had no effect at baseline (Figure S1 and S2). How do the authors explain these discrepancies?

We have employed the ion substitution experiments to mimic the primary cellular effects of electrical stimulation, the strong changes in intracellular concentrations of both Ca^2+^ and Na^+^. It is true that rapid changes might obey somewhat different rules than the slower changes we are able to make by ion exchange, but we think that the absence of a response to Ca^2+^ when Na^+^ is unavailable for exchange via NCX is quite informative. Combined with the observation in the previous paper showing that Ca^2+^ influx is normally an important contributor to the increased glycolysis during stimulation, we believe this makes a good case for our conclusions: that Na^+^ is required for converting the Ca^2+^ signal into energy demand and a glycolytic response. We also corroborated the NADH_CYT_ responses that we observed in our ion substitution experiments with the Ca^2+^ puff experiments to test the effect of a faster change in free cytoplasmic Ca^2+^, more similar to what is achieved through electrical stimulation. The amplitudes of the Ca^2+^ increases that we achieve with both the ion substitution experiments and Ca^2+^ puff experiments are also similar to the typical amplitudes of Ca^2+^ increases when evoked by electrical stimulation. Unfortunately, it is impossible to produce either synaptic or antidromic stimulation of the neurons in the absence of extracellular Na^+^.

We have now added a paragraph with this information to the Discussion.

The effect of thapsigargin in the context of electrical stimulation (shown in Figure 1 Supplement 2) is to increase the size of the intracellular [Ca^2+^] transients, and it also produces an increase in the Peredox response, normalized to the RCaMP response. As mentioned above (response #4), this can be explained if a larger than normal fraction of the intracellular Ca^2+^ is removed by NCX because of the absence of SERCA-mediated uptake into ER.

The finding that inhibition of SERCA during stimulation actually elevates cytosolic NADH level argues against Na^+^ being the only ion that regulates glycolysis.This is an interesting finding and there is no need to explain it away. In fact, it may be that it is local microdomains of Ca^2+^ that activate glycolysis (raise NADH level) rather than bulk Ca^2+^.

Our Ca^2+^ puff experiments show that large transient increases to Ca^2+^ that likely spread to all parts of the cell have minimal effect on NADH_CYT_ increases, though Ca^2+^ can certainly affect glycolysis by modulating the influx of Na^+^ through the NCX. The slight increase to the NADH_CYT_ transient after SERCA inhibition is likely because the ca^2+^ that is not taken up by ER must be extruded by the NCX, which would increase the ATP consumed by the Na^+^/K^+^ pump. We have added more clarity on this to the first and second sections of the Discussion.

The finding that "SBFI ΔF/F transients were longer in duration than the RCaMP LT transient" does not necessarily mean that Na^+^ elevation lasts longer than Ca^2+^ in the cell. This could be an artefact of the SBFI on/off rate relative to RCaMP. In fact, prolonged elevation of cytosolic Na^+^ would make neurons refractive to depolarization in AP trains.

It is highly unlikely that the prolonged Na^+^ transient we observe, lasting nearly a minute, is an artifact of SBFI on/off rates. The complexation of alkali metal cations by crown ether molecules is kinetically very fast. The rate of association of Na^+^ with crown ethers is in the range of 10^7^ – 10^10^ M^-1^ s-1 and the rate of dissociation is in the range 10^4^ – 10^6^ s-1 (Adamic et al. 1986, doi:10.1021/j100282a030 and Liesegang et al. 1977 doi:10.1021/ja00452a006). The relaxation time constant of Na^+^ with crown ether molecules, calculated using these rate constants with an intracellular [Na^+^] = 5 mM, is in the low microsecond range, meaning that it will achieve nearly its entire response in ≤ 50 μsec. We can expect the kinetics of Na^+^ complexation with SBFI, which is a crown ether with several fluorophores attached, to be similar. We have added these citations to the Methods section on single-cell electroporation and use of SBFI.

Prolonged elevation of cytosolic Na^+^ alone (to the levels seen here) should not cause neurons to be refractory to firing; refractoriness typically occurs in the setting of prolonged depolarization and consequent inactivation of NaV channels.

Also, RCaMP has a t_1/2_ decay of 410 ms (Akerboom et al. 2013 doi:10.3389/fnmol.2013.00002). We have added this citation to the last section of the Discussion.

To ensure that the interpretation of changes in cytosolic NADH are valid with respect to glycolysis, it is important to measure dynamic changes in other metabolites such as glucose.

Measurement of intracellular [glucose] would not help validate the conclusions or refine the interpretation of the NADH signal, since an increase in glycolytic rate is compatible with any change in the level of glucose: up, down, or no change. Levels of glucose and other metabolites are dependent on their own intrinsic, and variable, rates of import/production and export/consumption, so the interpretations of measuring the levels of additional metabolites on the rate of glycolysis is not straightforward. We have also previously reported two extensive characterizations showing the NADH_CYT_ signal originates from glycolysis (Diaz-Garcia et al. 2017 doi:10.1016/j.cmet.2017.06.021 and Diaz-Garcia et al. 2021 doi:10.7554/*eLife*.64821).

Pharmacological inhibition of the NCX transporter is needed. Many of the conclusions about its role are based on manipulation of extracellular Ca^2+^/Na^+^ alone.In addition to altering the ionic compositions, chemical blocking of NCX (SEA0400?) might be useful to validate the role of NCX in the proposed pathway.

Although there are many published inhibitors of NCX transport, we are not aware of any that is really effective at specifically inhibiting the forward transport mode of the transporter, which would be needed for informative results in our experiments. Most reported inhibitors of the NCX are only mildly effective at inhibiting the reverse mode but have other non-specific effects and variable isoform selectivity (Sharikabad et al. 1997 doi:10.1111/j.1600-0773.1997.tb00284.x, Iwamoto et al. 2006 doi:10.1124/mol.106.028464, Secondo et al. 2015 doi:10.1021/acschemneuro.5b00043).

SEA0400 does not effectively inhibit the forward transport mode of NCX (Iwamoto et al. 2004 doi: 10.1074/jbc.M310491200), and it affects Ca^2+^ channel function, so interpreting the results from stimulation in the presence of SEA0400 would also be complicated. We have now added a brief sentence about NCX pharmacology in the Discussion.

Monensin might be useful to show the effect of a fairly direct activation of the Na^+^/K^+^ pump on glycolytic activity.

Monensin is a Na^+^/H^+^ exchanger, so monensin-induced changes in Na^+^ would be accompanied by pH changes. This would complicate the interpretation of any monensin-induced effect on glycolysis, since the activities of several glycolytic enzymes are pH-sensitive.

Switching back from 0 Na^+^ + 0.5 Ca^2+^ to 147 Na^+^ + 0 Ca^2+^ led to a sustained NADHcyto increase. Cytolasmic [Na^+^] shows only a transient rise when Ca^2+^ entry is inhibited (Figure 4C). Please explain.

These are different experimental paradigms that cannot be compared directly. The switching from 0 Na^+^ + 0.5 Ca^2+^ to 147 Na^+^ + 0 Ca^2+^ is an external ion manipulation that is designed to evoke very long-lasting changes in intracellular ion concentrations, while the cytoplasmic [Na^+^] transients are recorded under a more physiological condition with briefer stimulation via an electrode.

Please show NADHcyto during antidromic stimulation in presence of Ca_v_ blocker.

This is shown and described in detail within our earlier report (Figure 3 of Diaz-Garcia et al. 2021 doi:10.7554/*eLife*.64821), which is cited in this report’s Discussion.

The cyto-Ca^2+^ response is lower after Ca^2+^ puff in 0 Na^+^ and with Ouabain treated condition. What could be the possible explanation?

These are not paired responses. The differences in Ca^2+^ transient amplitude are not due to applying ouabain or to removing Na^+^, but are simply due to the varied effectiveness of the Ca^2+^ puff during each recording. The scatter plot of the Ca^2+^ puff data shows a clear comparison of data across each condition for a wide range of Ca^2+^ increases.

We have added more clarity to the Results section to indicate that the Ca^2+^ puff recordings in different external conditions are not paired. It is also stated in the Figure 3 legend that the ∆Peredox/∆RCaMP ratio significance test is for unpaired samples using a Mann-Whitney test.